# Transforming brownfields into urban greenspaces: A working process for stakeholder analysis

**Shaswati Chowdhury**[1]*, **Jaan-Henrik Kain**[2], **Marco Adelfio**[1], **Yevheniya Volchko**[1], **Jenny Norrman**[1]

**1** Department of Architecture and Civil Engineering, Chalmers University of Technology, Gothenburg, Sweden, **2** Gothenburg Research Institute (GRI), The University of Gothenburg, Gothenburg, Sweden

* shaswati@chalmers.se

## Abstract

Urban greenspaces (UGS) provide a range of ecosystem services and are instrumental in ensuring the liveability of cities. Whilst incorporating UGS in increasingly denser cities is a challenge to planners, brownfields form a latent resource with the potential of being converted into UGS. Transformation of brownfields to greenspaces, however, requires engagement of a variety of stakeholders, from providers to users. The overall aim of this study was to support effective and realistic realisations of UGS in the context of urban brownfields' regeneration and stakeholder engagement. A working process was developed to: 1) integrate methods relevant for UGS realisation for a) identification and categorising of relevant stakeholders, b) mapping their interests and resources, c) identifying various challenges, and d) matching those challenges with the mapped resources over the timeline of UGS development; and 2) apply these methods to assess relevance and shortcomings. The methods were applied to a study site in Sweden, and data was collected using a questionnaire survey. The survey received 31 responses and the respondents' comments indicated that the combination of several uses, especially integrated with an urban park, is preferable. Visualisation was an important component for data analysis: stakeholder categorisation was effectively visualised using a Venn diagram, and the needed mobilisation of resources among stakeholders to manage identified challenges was visualised using a timeline. The analysis demonstrates the need for collaboration between stakeholders to achieve an effective realisation of UGS and how multiple methods can be used in concert to map stakeholders, preferences, challenges, and resources for a particular site. The application at a study site provided site-specific data but the developed stakeholder categorisation, and the method for matching identified challenges with the stakeholders' resources using a timeline, can be generalised to applications at other sites.

**Data Availability Statement:** All relevant data are within the paper and its Supporting information files. The meta data is available by the following

link: https://data.qdr.syr.edu/dataset.xhtml?persistentId=doi:10.5064/F6UB6BBL.

**Funding:** This work is supported by Formas (Grant number: 2017-00246).

**Competing interests:** The authors have declared that no competing interests exist.

## 1. Introduction

Cities largely consist of built-up areas, but pockets of green open spaces can be found scattered in the urban fabric. Such vegetated Urban Greenspace(s) (UGS) have proved to be essential for the physical and mental well-being of the inhabitants as well as providing a multitude of ecological functions [1–5]. The importance of UGS as an indicator of liveability came to prominence during the twentieth century when cities started to expand exponentially [6]. Half of the world's population is now urbanised and very soon cities will represent the larger share of the global demographic, projected to increase up to 60% of the global population by 2030 [7]. As the city-dwelling population continues to rise, the pressure to develop greenspaces in urban areas is increasing [4, 8]. Furthermore, inadequate funding, lack of maintenance and increased privatisation of greenspace management make UGS increasingly vulnerable to urban encroachment [9]. The COVID-19 pandemic has limited the mobility of urban inhabitants and has made UGS an essential, and often the only, accessible outdoor element of everyday city life [10, 11]. Therefore, this is an element to take into account in urban planning in preparation towards future uncertainties. A sustained increase in the number of users and intensity of use would require an increase in both the number and the variety of UGS [10, 12, 13]. Even before the COVID-19 pandemic, many UGS in cities were already encapsulated in densely inhabited built-up areas which exacerbated their intensity of use and in some cases even undermined their existence under the pressure of market economy and speculation [9]. The compact city is the by-product of the sustainable planning approach for urban development that has pushed for more densified cities to tackle urban sprawl [13].

To meet the growing demand for UGS in cities, urban land is needed. As land in the city is a scarce resource, the once used but currently abandoned parcels of land scattered across cities, known as brownfields, form a latent resource that holds the potential of being used for UGS. Largely due to the previous use, brownfields are potentially contaminated but are usually supported by existing infrastructure. They are often the only available option for redevelopment in densely populated cities [14]. Brownfields in their present state can be considered as an UGS, 'abandoned/ruderal land' that supports spontaneous vegetation based on the UGS typology by Haase et al. [15]. Though brownfields can provide some regulating and supporting ecosystem services in their present state, their use is unregulated and limited largely due to potential contamination from previous exploitations [16, 17]. Retrofitting brownfields as more usable greenspaces can add on to the cultural ecosystem services as well as increasing the environmental benefits [17]. In Sweden alone, there are approximately 85 000 potentially contaminated sites [18] and the corresponding number for Europe is approximately 2.5 million [19]. Bringing obsolete land back in use is an essential part of circular economy adaptation in cities [20]. Rather than being the waste of the linear land use system, brownfields are a valuable resource that presents the redevelopment opportunities in a circular urban land use strategy [20–22]. Restoration of brownfields is key in the EU's 'No net land take by 2050' policy [23] and in the EU soil strategy [24].

Apart from the recent policy push, one driving factor in brownfield remediation in Europe is the prospective market value of the reclaimed land [25]. The anticipated profit (expected return on investment) often drives the redevelopment of brownfields into residential, commercial, or industrial uses [26]. UGS uses provide a myriad of non-market ecosystem services that is not necessarily fully reflected in the market value of land. Attractive urban brownfields have therefore a higher chance of redevelopment, whereas financially unattractive sites may remain abandoned for a long time [27]. Still, even before the pandemic-driven boost in demand for UGS, over 19% of the redeveloped brownfields were retrofitted with greenspaces in the UK from 1988 to 1993 [26]. To support transition of urban

brownfields to UGS, Chowdhury et al. [21] suggested a bio-based land use matrix for a tentative selection of UGS and provided a description of ecosystem services delivered by these UGS. Potentially, a wide range of stakeholders will be involved in greening the brownfields due to their diverse interests in ecosystem services, and any initiative to redevelop brownfields into UGS would need planning processes involving considerate and consistent facilitation of multi-stakeholder engagement.

For this study, stakeholders are defined as "any individuals or groups who affect the project, or are affected by it, or exhibit an interest in it" [28, p. 7]. Stakeholders across different domains, such as public (e.g. local and national governments) private (e.g. enterprises and businesses), and society (e.g. residents, community-based organisations and academic institutions) play important roles in UGS development and performance in general [29]. For developing UGS on brownfields in particular, the importance of government stakeholders was further confirmed by [26], who analysed 12 such examples in Toronto (Canada) and found all redevelopment projects were carried out by the public sector, with the majority being overseen by the municipal parks department. Participation of society in the planning process can also ensure a consistent and conscious future use of UGS by creating collective awareness [29, 30]. Furthermore, public involvement in UGS development and management leads to legitimate and informed decision making, whilst increasing sense of ownership, sense of belonging, and a willingness to maintain UGS [31, 32]. However, conflicts of interests among some stakeholder groups, as well as mismanagement of their resources, are also relatively common, and the increased involvement of NGOs in public space development across Europe and the US can be attributed to a public dissatisfaction with regulatory planning failures [29]. Particularly for the US, neo-liberal policy making and practice have reduced public sectors' engagement in the generation of green spaces contributing to their privatisation and commodification [33, 34]. Developing UGS can, hence, potentially become a long and arduous process if a proper method for identifying and involving stakeholders is not implemented at an early stage [35]. A comprehensive understanding of relevant stakeholders' preferences, needs, values, knowledge, and the resources they could bring is, therefore, required to ensure a well-informed decision on the redevelopment of brownfields into certain UGS.

The overall aim of this study was to support effective and realistic realisations of UGS in the context of urban brownfields' regeneration and stakeholder engagement. To achieve this overall aim, a working process was developed to:

1. integrate various methods relevant for UGS realisation, including methods for a) identification and categorisation of relevant stakeholders, b) mapping their interests and resources, c) identifying the various challenges associated with realising UGS on urban brownfields, and d) matching those challenges with the mapped resources over the timeline of UGS development; and

2. apply methods within this working process to assess its relevance and identify potential shortcomings.

The work is demonstrated at the study site Polstjärnegatan, situated in Gothenburg, South-West Sweden.

## 2. Methodology

The working process of this study included a sequence of steps, Fig 1. Section 2.1 describes three selected methods for a) identification and categorising of relevant stakeholders, b) mapping interests and resources, c) identifying the various challenges. Section 2.2 describes the development of a new method (1d in Fig 1) that combined the output of the selected methods

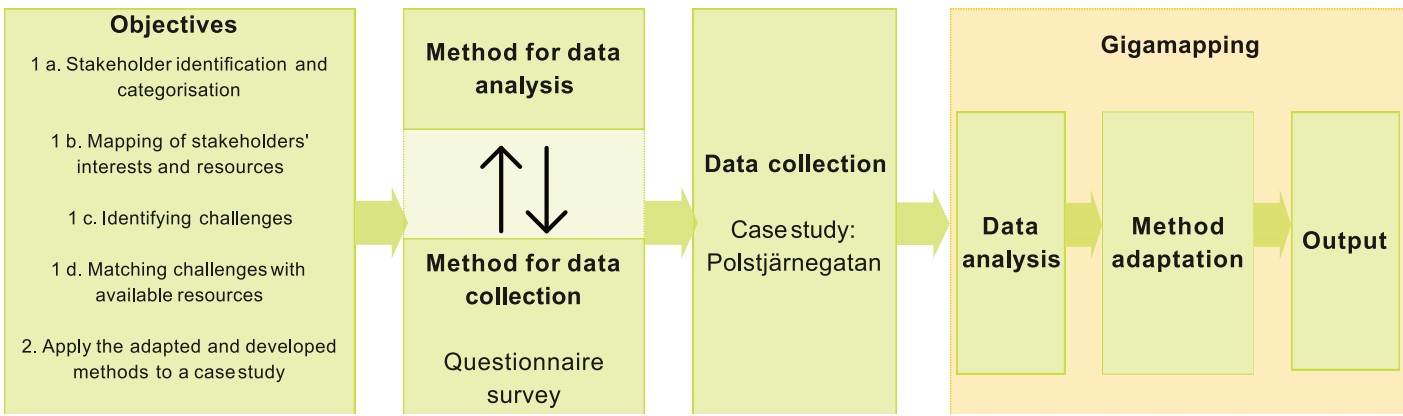

**Fig 1. An overview of the working process used in this study.**

to match challenges with resources. A method for data collection is described in Section 2.2. The study site Polstjärnegatan which was used for data collection and analysis is described in Section 2.3.

Gigamapping [36, 37] was used for assessing, adapting, and applying the selected methods for stakeholder analyses, and also for structuring and analysing the collected data. Gigamaps were produced using Miro (miro.com), an online platform for visual mapping and collaboration. Miro's web platform was used by the first author to process the data and to document the analyses of multiple processes with the aim of co-discussing them with the other authors using the interlinked diagrams and visualisations. An advantage of Gigamapping is that it separates analytical tasks and communication tasks in the mapping process. This gives the freedom to create complex initial mappings, which can then be gradually simplified and refined until a final output is derived, one that is possible to communicate externally [36]. However, as Jones and Bowes [38] explained, Gigamaps also present complex social and design challenges and are, thus, intentionally dense and multi-levelled, and therefore often challenging to produce. The full complexity of the final Gigamap produced in this study can be appreciated in the attached S1 File.

## 2.1 Methods for stakeholder analyses

Three methods for stakeholder analyses were selected from the available literature using a scoping review [39] (see S1 File). Two selection criteria were used: i) contextual similarity and ii) ease of application for data collection. These three selected methods were: a) identifying and categorising relevant stakeholders [40], b) mapping the interests and resources of these stakeholders [41], and c) mapping the various challenges associated with realising UGS on urban brownfields [42].

For stakeholder identification and categorisation, the categorisation by Rizzo et al. [40] was selected because of the similarity in the premise of brownfields in both studies. Following Rizzo et al. [40], a generic list of ten sub-groups of stakeholders was created for this study: (1) site owners, (2) local/city government, (3) regional/national government, (4) controlling authorities, (5) landowners of the surroundings, (6) local business/other activities, (7) civil society/NGOs/not-for-profit organisations, (8) local residents, users of the greenspace today, (9) site developers, and (10) others. This list was further used for collection of site-specific data with the questionnaire (Section 2.2), where the respondents categorised themselves into one or

more of these ten stakeholder sub-groups, as well as provided more information about their position and organization while staying anonymous if desired. The respondents were also asked to identify other relevant stakeholders associated with their preferred UGS.

For analysing stakeholders' interest and resources, the Crosby method [41] was selected to be tested in the study due to the relative ease of application and the potential for modifying it to the needs of a specific study. This approach uses a matrix to analyse stakeholder groups according to (i) the group's interests, (ii) the level of resources it possesses, (iii) its capacity for mobilisation of resources, and (iv) the group's position on the issue in question [41, 43]. The standard data collection method for stakeholder analysis with the Crosby method uses personal interviews but informal panel groups and workshops have also proved to be successful [41]. However, due to COVID-19 such data collection methods were impossible to be used in this study. This led to focusing only on (i) stakeholders' interests and (ii) their resources. An 'interest' was interpreted as a stakeholders' preference for an UGS to be developed on the study site which is presently in derelict condition with sparse vegetation (more detail on section 2.4). 'Resources' were explained as the resources required for developing green land uses, with three examples provided: time, money, and knowledge. The explanation of resources was left open for stakeholders' own interpretation encouraging them to elaborate on the resources they could contribute with, to realise their preferred UGS.

To identify and analyse challenges, the challenge categorisation proposed by Fernandes et al. [42] was selected due to the contextual similarity (brownfield development) and because it was relatively easy to categorise the collected data. These categories included governance, infrastructure, territory, finance, culture, and environment. The stakeholders were asked to identify and describe challenges to develop the studied site in line with their preferred UGS.

## 2.2 Method for matching challenges with resources

A new method was developed to match the identified challenges with the mapped resources over the timeline of UGS development. The method was developed with help of Gigamapping on the Miro visual platform using the data derived from application of the three above-described methods at the study site.

## 2.3 Data collection

Two alternatives for data collection were initially considered: i) a focus group discussion followed by in-depth interviews and ii) a questionnaire survey followed by in-depth interviews. However, due to the COVID-19 pandemic and the associated restrictions, any prolonged in-person procedures were deemed unsuitable. Instead, a digital questionnaire survey in combination with an in-person survey version were selected for this study. A questionnaire survey for data collection also meant that respondents could answer at their own convenience and being largely online and anonymous, helped the respondents to be frank with their answers [44]. The online survey platform Momentive (https://www.surveymonkey.com/) was used to carry out a digital questionnaire survey during spring and early summer 2021.

The bilingual questionnaire (Swedish and English) was designed to capture the UGS preferences of local stakeholders and what resources and challenges they saw in connection with these UGS. The English version of the questionnaire is provided as S2 File. The three selected methods for stakeholder analyses (Section 2.1) helped to formulate the questions, to collect appropriate site-specific data, and to understand the applicability of the methods. In the questionnaire, respondents were first asked to select three UGS from a list of ten, with one option being 'other' where the respondents could include an UGS that they preferred but which was not on the list. The list of possible UGS at the study site and description of these UGS in the

questionnaire was based on Chowdhury et al. [21], using the classification/categorisation of the Greensurge project [15]. The UGS presented in the questionnaire were: bioswale, urban park, grassland/shrubland, meadow orchard, allotment, community garden, commercial agriculture, biofuel production, and spontaneous vegetation. The questionnaire had both close-ended (e.g. selection of UGS from a list) and open-ended questions (e.g. explaining the reasons behind UGS preferences).

Local stakeholders were initially identified using the Rizzo et al. [40] categorisation and the data collection process started with the members of a reference group to the research project. The online survey was first distributed via email to the reference group members, who were mostly municipality representatives. They were asked to answer the questionnaire and forward it to their colleagues and anyone potentially knowledgeable about the survey topic. In this way, a snowball effect of identifying stakeholders and survey respondents was expected. In addition, other potential local stakeholders were identified by the authors and invited to participate in the survey via email. To ensure public participation, posters were created in both Swedish and English with a QR code to access the survey. Printed versions of the poster were put up on notice boards in the local student housing. Digital versions of the poster were displayed on screens located at the Chalmers University of Technology Campus in Lindholmen, Gothenburg, located in the proximity of the study site. The digital posters were displayed from 26th to 30th April. The weblink and email invitation for the survey were active from 2nd March, and the QR code from 14th April. The links were closed on July 19th. Also, some in-person surveys were carried out around the study site on 13th April.

In total, 31 survey responses were collected. Twenty-six of these were responses on the digital platform (Momentive) and five were responses collected in person near the site. The survey completion rate among respondents was 97%, meaning very few respondents skipped questions and, on average, a respondent spent 11.5 minutes on the online version of the survey. Fifteen respondents preferred to stay anonymous whereas sixteen offered to provide additional information if needed. The most represented stakeholder group was local resident (18) followed by local/city government (6). Regional/national governments were represented by two respondents while one respondent identified themselves as a site owner. An ethical review was not required as the content (no sensitive personal data) of the questionnaire survey for this study is exempt from any such requirement according to the Swedish data protection act [45]. The online questionnaire survey was performed with explicit consent of the participants as this was required to be able to respond to the survey. For offline data collection, the respondents were first asked to consent to the purpose of the study and the survey was filled only after the verbal consent was given. Both online and offline survey participants could choose to stay anonymous.

## 2.4 Study site

The study site at Polstjärnegatan is located within the Lindholmen district in Gothenburg (Sweden). Population of Lindholmen is around 5000 according to 2022 census but more than 10000 students come to study here due to the district being an education hub with several schools and a campus of Chalmers University of Technology [46, 47]. Lindholmen district has less than 3000 housing units at present and a significant part of them are student apartments (nearly 700 registered apartments by the student housing associations alone) [46, 48]. The site is part of the concept plan of Karlastaden, a large-scale housing and commercial facility (Fig 2, top), whose construction (as of 2022) is in progress [49, 50]. The development is projected to finish by 2026 and will add 2000 more apartments to the Lindholmen area [49, 50]. The site has confirmed contamination issues due to previous uses [51]. It is surrounded by roads and a

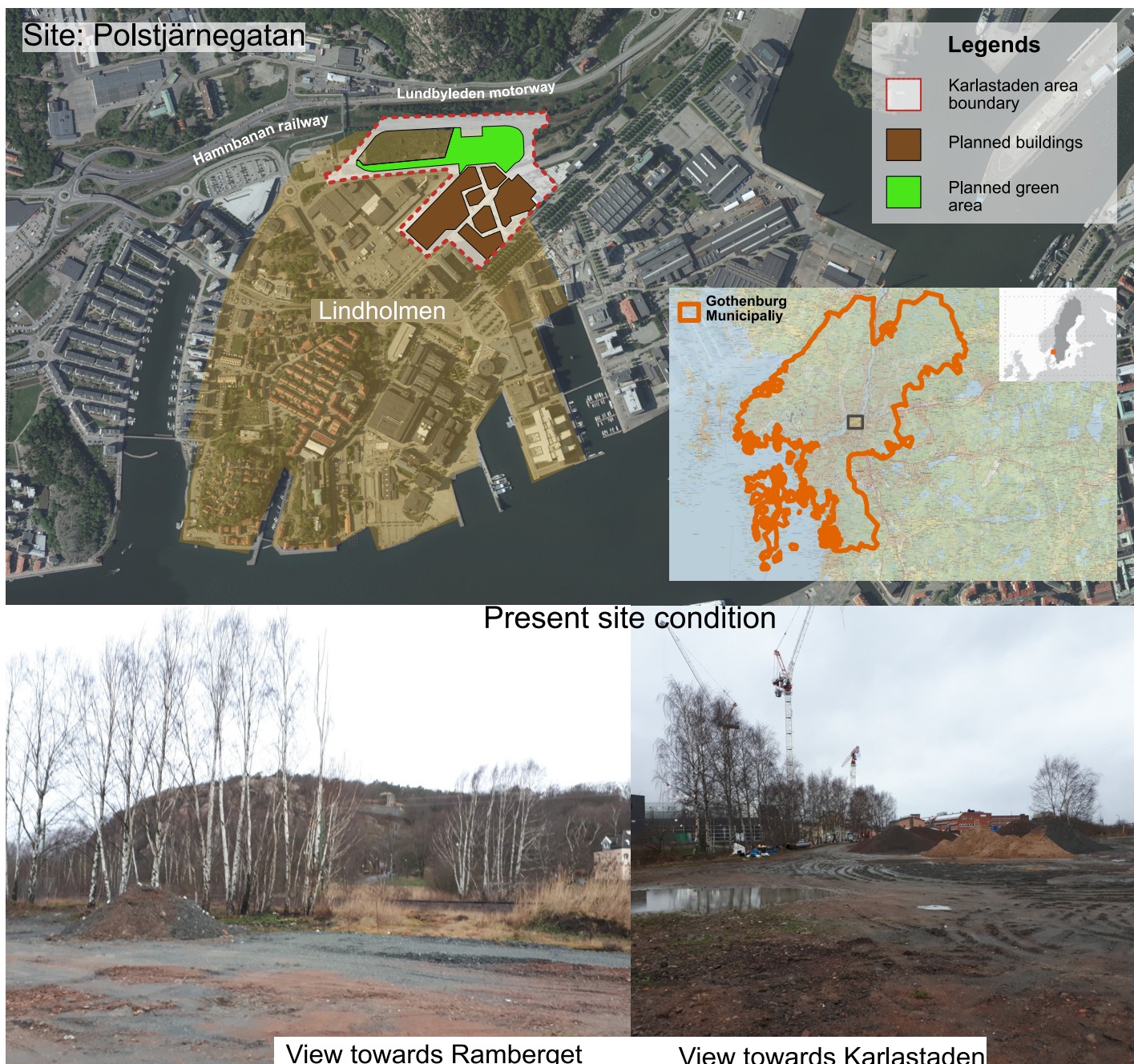

**Fig 2.** Top: Location map integrated with the concept plan of the Karlastaden development project with the site area highlighted (Karlastaden plan redrawn from Göteborg stad [53]; base map and orthophoto ©Lantmäteriet (https://www.lantmateriet.se/en/maps-and-geographic-information/maps/), used under Creative Commons License CC BY 4.0). Bottom: Physical conditions of the site as of January 11, 2020 (source: first author).

railway on all sides: Polstjärnegatan to the south, Karlavagnsgatan to the east, a petrol station and a fast-food restaurant to the west, and a railway (Hamnbanan) as well as a motorway (Lundbyleden) to the north (see Fig 2).

The current planned future use of the site is as a park area, specially designed to help with the surface water runoff with new roads being constructed around its perimeter. The study

area comprises about 14,800 m$^2$, mostly consisting of sparsely vegetated spaces at present (Fig 2, bottom). Part of the site is currently being used to store construction materials and construction sheds for the workers involved in building work at Karlastaden. The site is being rented by the construction company Serneke for the duration of the construction period.

Historically, most of the site was used as a yard for coal products, forming part of Sannegårdshamnen and its shipyard. The shipyard was in operation from the early 1900s to the 1980s. After the shipyard was closed, the site was turned into a golf course, demolishing the yard structures and the old railroad, with sludge brought from Ryaverket (a sewage treatment plant) to model the surface. The golf course was closed in the early 2000s and, since then, the site has remained unused for the most part. However, several illegal cable burning areas for metal reclamation have been found which are now acting as small hot spots of metal and PCB contamination. For detailed information regarding the contamination situation at the Polstjärnegatan site, see Chowdhury [52, pp. 41–43].

## 3. Results

### 3.1 Identification and categorisation of stakeholders

All the stakeholders relevant for UGS realisation identified in the Polstjärnegatan, could not be categorised using the selected method by [40] because a) several additional stakeholder types were identified by the survey respondents, and b) the new stakeholder types identified by the respondents either belonged to several stakeholder categories or could not be characterised well within the original categories. Consequently, the applied ten categories by Rizzo et al. [40] were not fully able to capture and describe the relevant stakeholders and their role relating to the realisation of UGS in the study area.

Following the responses from the questionnaire survey, an interconnected and flexible stakeholder categorisation is proposed, where the diverse roles of some stakeholders are recognised by allowing them to be represented in more than one stakeholder group. The proposed categorisation has one main category (*everyone*) which then includes three cluster categories (*government*, *local community*, *non-local community/visitors*), where stakeholders can appear in more than one group. All stakeholders relevant for UGS realisation have been further categorised into groups or sub-groups within the main cluster categories. The developed stakeholder categorisation is demonstrated by a Venn diagram (see Fig 3). It shows the main category, the three generic cluster categories and the site-specific stakeholder groups identified in the study. The largest group of stakeholders that encompasses all the other subgroups of stakeholders is *everyone*, representing all members of society. Terms such as 'all', 'all stakeholders', 'the public', 'all really, children and adults' and 'anyone, everybody should be' are used ten times by six respondents for five different UGS options (responses were given both in Swedish and English, and the Swedish responses have been translated into English by the authors). Every member of the society could potentially benefit from realising UGS and could, therefore, be considered as a relevant stakeholder for realising UGS.

The cluster category *Government* consists of two stakeholder groups, *municipal/local government*, and *regional government*. The municipal government is perceived to have a stronger role to play as they are mentioned using various terminologies by sixteen respondents; six of these respondents represent the local government themselves. Several departments within the municipality that can play specific roles in UGS realisation are identified by the survey respondents: the Parks and Nature Administration (Park och Naturförvaltningen), the Recycling and Water Office (Kretslopp och vatten), the Environmental Administration (Miljöförvaltningen), the City Planning Authority, (Stadsbyggnadskontoret), the Real Estate Office (Fastighetskontoret), and the municipal urban development company Älvstranden Utveckling AB. Unlike

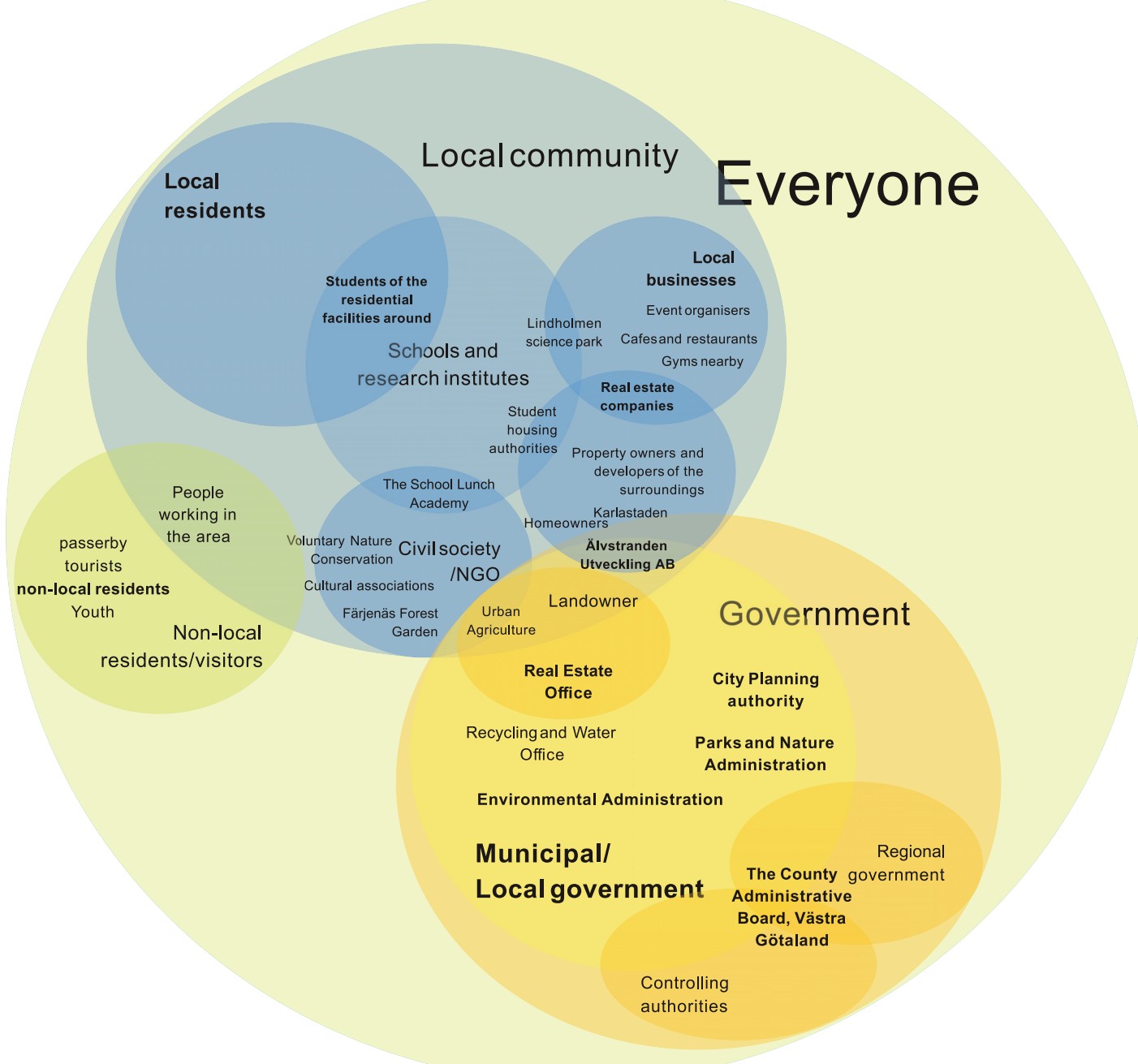

**Fig 3. Mapping of identified and categorised stakeholders at Polstjärngatan.** * The stakeholder groups represented by the survey respondents are in bold.

the local government, the regional government is not singled out by any respondent as a stakeholder that could be interested in the issue of realising UGS, not even by the two respondents who identified themselves as being its representatives. Still, some departments of the regional government, along with some branches of the local government, form the *controlling authority* group that is likely to play a role in, for example, local and regional land use and safety regulations.

The *local community* cluster category consists of six stakeholder groups, including *landowner* (of the site), *property owners and managers of the surroundings*, *civil society/NGOs*, *local residents*, *schools and research institutes*, and *local business*. These groups often overlap and are interlinked with some stakeholders that belong to more than one group or category. Eighteen out of thirty-one respondents are local residents, and almost all respondents identify local residents as relevant stakeholders. As the study site is located near a university campus with several student housing blocks nearby, students living in these housing facilities are also specifically mentioned as local residents that could benefit from realising UGS on the study site. Students living in the residential facilities nearby are also part of the stakeholder group *schools and research institutes*. The Lindholmen Science Park, a collaborative platform between the business community, university, and the municipality, connects schools and research institutes with local business. Not many local businesses are identified but the few mentioned are event organisers, cafés and restaurants, and local gyms. One particular type of business mentioned frequently is real estate, with many such businesses operating in the vicinity of the site. They also overlap with the stakeholder group *property owners*, *developers and managers of the surroundings* which is frequently identified. In total, members of this group are mentioned fourteen times by eight survey respondents. Two student housing authorities represent property owners as part of an institute or municipality. Thus, they both represent *property owners and manager of the surroundings* and *schools and research institutes*.

The *government* cluster category includes five stakeholder groups: landowner, real estate companies, the municipal local government, the regional government and controlling authorities. The current landowner of the study site is the municipality represented by the Real Estate Office, but the municipality also owns and represents many other properties in the surroundings. The municipality operates through Älvstranden Utveckling AB, responsible for developing the waterfront of the city. Several developers and, in part, also the municipality, are responsible for developing the mixed urban development project Karlastaden, of which the study site is part. Stakeholders representing *civil society/NGOs* are rarely mentioned, but some are identified for specific UGS: the association Voluntary Nature Conservation (Ideell naturvård) is suggested as being interested in realising natural UGS, such as grassland or shrubland; Färjenäs Forest Garden (Färjenäs skogsträdgård) is a nature conservation NGO working to protect local pollinators and might be interested in the development of a meadow orchard; the association Urban Agriculture (Stadsnära odling) maintains municipal allotment plots and is identified as a stakeholder relevant for realising allotments and a community garden. Finally, the School Food Academy (Skolmatskadamien) is a regional network in the Västra Götaland region promoting good meals and good eating habits in schools, and is mentioned as a potential stakeholder by a respondent if the site is designed with an UGS that works or integrates collaborative school garden for nearby schools and kindergartens.

*Non-local residents and visitors* is the last cluster category. Passers-by, visitors, office workers, and youth are all part of this group and are mentioned occasionally as relevant stakeholders, in total nine times by five survey respondents. People working in the area are, however, regularly mentioned as an important stakeholder group. They are also part of the stakeholder subcategory *local community* due the frequency and extent of time they spend in the locality.

## 3.2 Mapping of stakeholders' interests and resources

The data required to fill in the Crosby matrix [41] were simplified for the study application due to the limitations of the chosen data collection method. The Crosby method uses a matrix that lists the stakeholder characteristics (e.g. interest, position, resources) as a necessary first step for the mapping, but the matrix itself does not provide any comparison between different

stakeholders based on their interests or resources. Instead, the graphical Gigamapping approach was used to analyse and present stakeholders' interests and resources which otherwise could not be easily visualised by using the Crosby matrix. In particular, the adapted approach in this study focuses on visualising which stakeholders, and how many of them, express similar interests and resources (see Figs 4 and 5). As a result, it becomes possible to differentiate between stakeholders' interests and resources, and their preferences could be highlighted and interpreted in various ways, such as number of mentions, type of mentions, or mentions by whom.

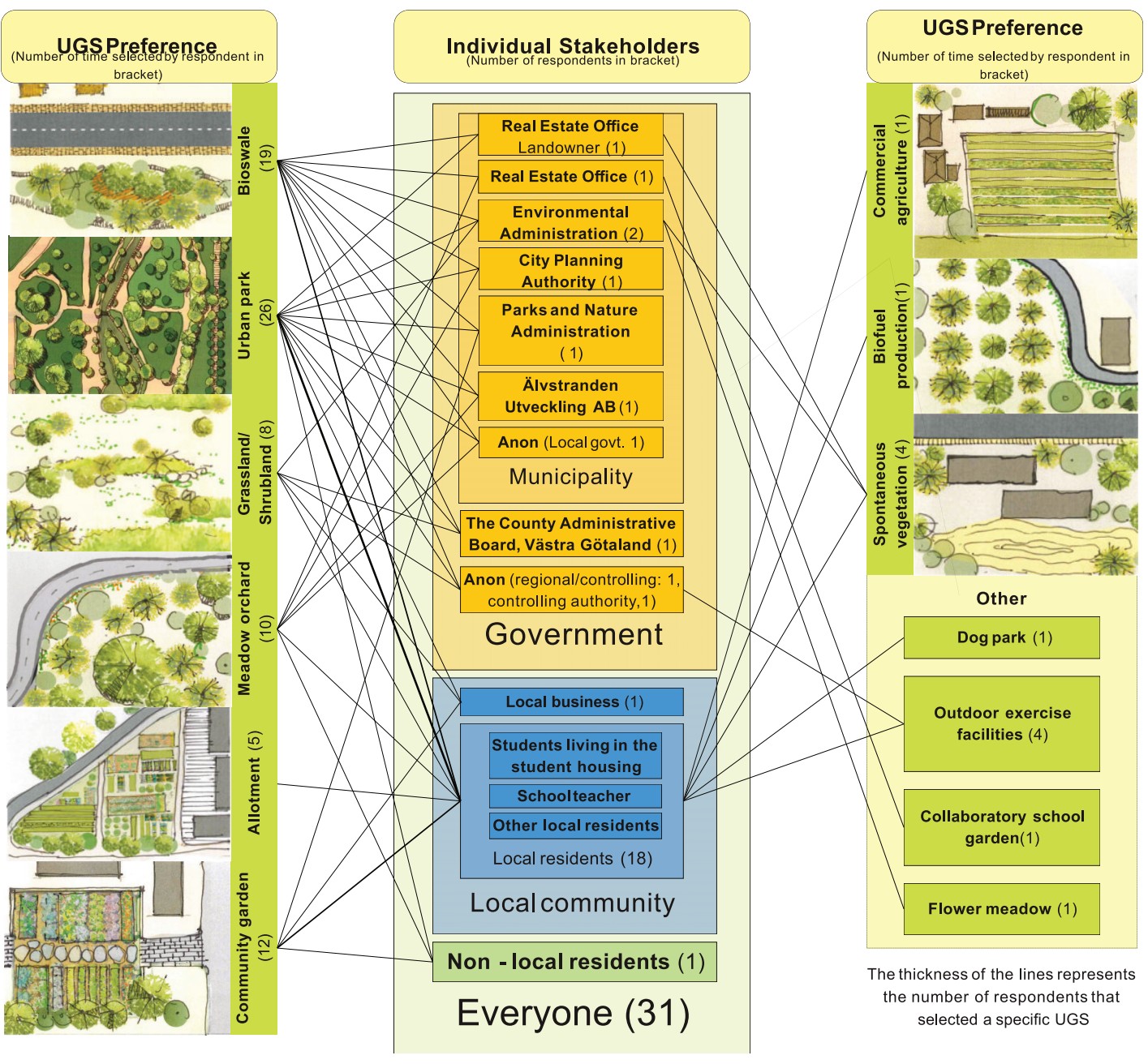

**Fig 4. Site-specific UGS preferences.**

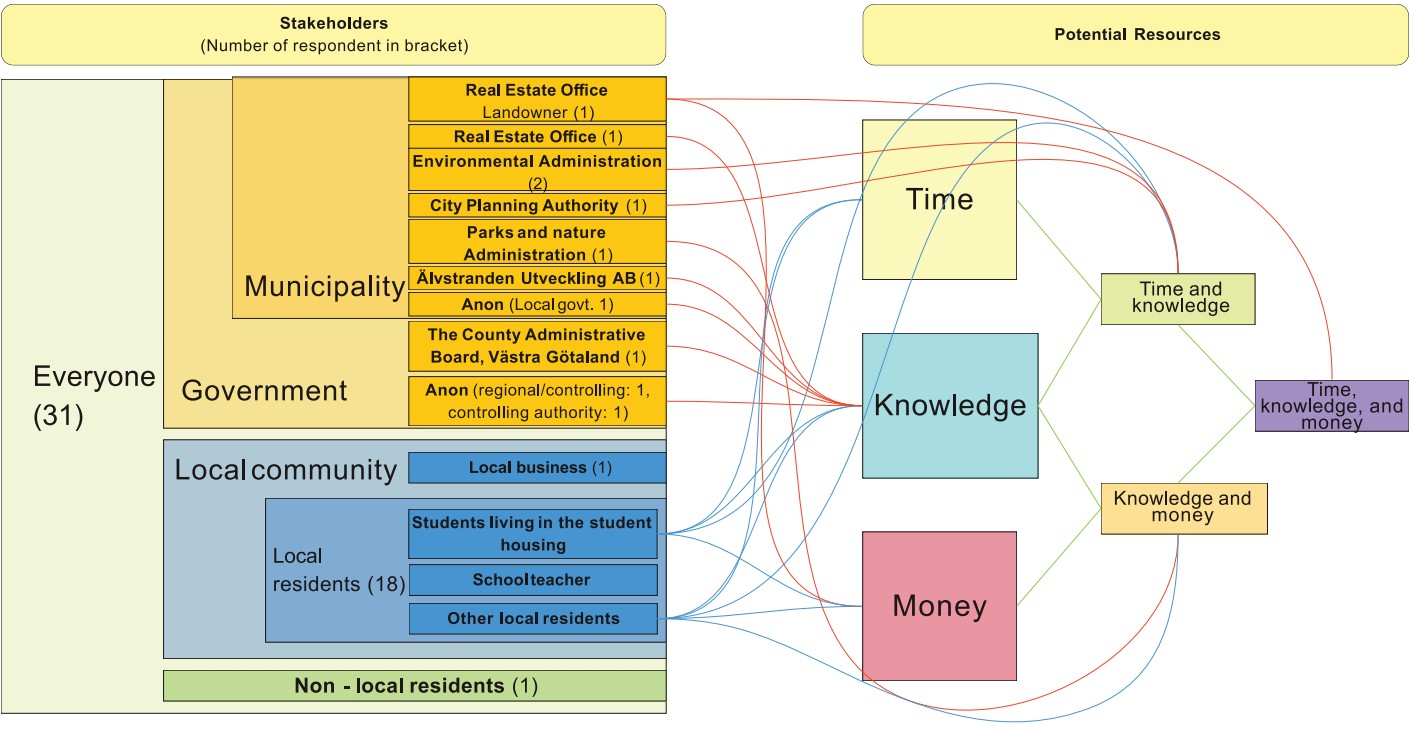

The thickness of the lines represents the number of respondents that selected a specific UGS.

**Fig 5. Stakeholders' potential resources at the study site.**

**3.2.1 Stakeholders' interests at Polstjärnegatan.** *Urban park* is the most popular choice of greenspace among the stakeholders, where 26 out of the 31 respondents selected this as a potential future use for the site (see Fig 4). The general reasoning among respondents for this preference is that people always need more parks in cities and that all the locals, especially the elderly and children, appreciate having an open recreational space. Respondents who are more knowledgeable about the site surroundings explain that there is a lack of parks in the area and that a park by the roadside would provide a place to rest and an opportunity for a brisk walk. The environmental and wellbeing benefits of parks are also mentioned by several respondents. The respondent from the Parks and Nature Administration also adds that larger parks offer more variety of park environments and thus provides the potential to include different activities. Several specific uses for parks are also identified. Four respondents say they would like to see an outdoor exercise facility, with one respondent suggesting the specific examples such as a skate park or an obstacle course. Another specific park use identified by the respondents is to include a fenced area for a dog park.

A *bioswale* is defined in the questionnaire as a vegetated and gently sloped pit for filtering surface water runoff (i.e. stormwater): 19 respondents selected this as a preferred potential future use of the land. In fact, all representatives from the municipality selected bioswale as a potential future use. Respondents who are knowledgeable of the site conditions insist on the necessity of bioswales in this area and several respondents stress the necessity of greenspaces to manage stormwater in urban areas in terms of both quantity and quality. The site has both a road and a railway along its northern edge and the surroundings mostly consist of hard surfaces. One respondent also describes how the sub-aquatic environments of the bioswale can be combined with vegetation to support various plant and insect species. Bioswales are described

by one respondent as nature-based solutions for storing and purifying stormwater as well as providing a range of ecosystem services such as supporting biological diversity. Such green solutions can contribute to build a city for the future.

Urban greenspaces that mimic more natural sites, here *meadow orchard*, *grassland/shrubland* and *spontaneous vegetation*, are also relatively favoured by stakeholders from both government and local community. Meadow orchard, consisting of fruits and berry shrubs and trees are a preferred UGS by ten respondents, belonging to six different stakeholder groups. The respondents mention that the provision of fruit trees would add quality to the recreational benefits of a park area, not only for people but also for birds and other animals. One respondent also describes a specific preference for a flower meadow that could also support endangered pollinators. Grassland/shrubland and spontaneous vegetation are selected as preferred future uses of the land by eight and four respondents, respectively. The respondents argue for the importance of such natural ruderal landscapes since they are, or can be, valuable biotopes for local flora and fauna that may be impossible to support in manicured greenspaces.

UGS that correspond to urban agriculture practices, *allotment* and *community garden*, are favoured specifically by respondents representing the local community. Allotment, that is, a collection of rentable small parcels of land, is one type chosen by five local resident respondents. Allotments in Gothenburg are currently in high demand and respondents mention the lack of them in the neighbourhood with a long waiting time for allocation. The respondents' reasons for preferring allotment over other uses include "peace of mind" and "calmness of the practice". Community garden is described as a communal agriculture and recreational space with the communal aspect of the use being specifically appealing to the respondents. Twelve respondents select this type of greenspace, describing it as a good initiative that would bring the community together with the help of collaborative activity. One respondent explains that it would be a rewarding and fun opportunity to grow and eat together with others in the community. Another respondent feels that such a greenspace would create an appealing neighbourhood by bringing people together who share a common interest, more so if visitors were also welcomed. A communal agriculture practice is also perceived by one respondent as an opportunity to educate citizens, especially children without access to gardening, about how food is grown. Another specific type of UGS is a collaborative school garden for nearby schools and kindergarten, named "co-laboratory" by one respondent who is interested in the education of children. Another benefit of engaging members of the local community in a community garden, according to one respondent, is that it could ensure better maintenance and, in the long run, increased security at the site.

*Commercial agriculture* and *biofuel production* are only selected once, but by two different respondents who prefer one of these two UGS. The respondent selecting biofuel production as an alternative is critical of the size of the site which is probably too small for such practices; they also feel that it is vital to consider different green solutions.

**3.2.2 Stakeholders' resources at Polstjärnegatan.** Stakeholders and the potential resources (*time*, *knowledge*, *money*, and combinations thereof) they could contribute to develop the study site as UGS are shown in Fig 5. The Real Estate Office respondent, representing the landowner of the study site, lists *time*, *knowledge*, and *money* as their potential resources. The most crucial resource is money, as the Real Estate Office is the main source of finance for developing the site. As the respondent representing the Parks and Nature Administration explains: 'As this is a development project, the financing and the money are with the Real Estate Office'. Respondents from other departments of the municipality that could potentially be involved in developing the site as an UGS (such as the Parks and Nature Administration, the Environmental Administration and the City Planning Authority) list *time* and *knowledge* as their resources. For municipal departments, *time* corresponds to 'human

resources in the form of working hours', and *knowledge* can include diverse areas such as 'expertise, planning documentation, project management, construction management, management of the completed park, access to network of competent designers, etc.'. Some departments of the local government that would not be directly involved in the site development could also potentially provide specific and in-depth *knowledge*. Älvstranden Utveckling AB, for example, has experience in developing other parks in the surrounding neighbourhood, such as the Jubileumsparken park in the Frihamnen area. Though there is no respondent representing the Water and Waste Recycling Department in the survey, this agency is identified by several respondents as a potential stakeholder for realising a bioswale, as stormwater management is their responsibility. The respondents representing regional authorities also list *knowledge* as a main type of resource.

*Time* is the resource most frequently mentioned by the local community. The interpretation of time as a resource for local residents is mostly as users of greenspaces. As one respondent explains: 'as a local not much maybe but will use the space when done'. Local residents can also be important sources of local *knowledge* crucial for the early design process. For uses, such as allotment or community gardens, three local residents who practise or previously have practised such uses, consider their experience as knowledge. They could share important information on local needs and issues if these UGS are to be realised on the study site. Three students living in the local housing list knowledge as a resource for realising *urban park*, with one respondent indicating an interest for further 'participation in the design of its function'. Local residents also mention *money* as a potential resource for UGS related to urban agriculture, such as allotment and community garden, by paying rent or fees.

### 3.3 Identifying and categorising challenges

The challenge categorisation proposed by Fernandes et al. [42] to map stakeholders' perception on challenges on brownfield redeployment is adjusted to address the specific challenges identified by the respondents. The suggested adjusted set of categories for categorising challenges for realising greenspaces on brownfields are: *governance*–includes issues that fall under the domain of government agencies, both municipal and regional, and also political visions; *land*–includes challenges regarding the location and size of the site, its accessibility, existing and planned physical facilities, restrictions such as urban zone mapping; *finance*–covers challenges involving solvency, and availability as well as access to financial resources, both public and private; *design*–includes challenges concerning the design of any future greenspaces at the site and ensuring the realised UGS is both aesthetic and functional; *sustainability*–includes a broad range of environmental and socio-economical concerns that can challenge the realisation of UGS and the access to the long-term benefits they provide. The modified categorisation of challenges was applied and visualised in the study (see Fig 6) and the results are described below.

The *governance* issues below, identified by the respondents, can occur at various stages of developing UGS at the site. At the early stages of design, it can be challenging to co-ordinate the planning process between the different departments of the municipal and regional governments. Even when a decision is taken and a plan is finalised for a greenspace to be realised, the speed of development is a concern that can arise, as addressed by one respondent, due to many construction projects going on in the surrounding area at the same time. After the UGS has been developed, ensuring maintenance of the greenspace (e.g. coping with a large number of visitors, damage over time, littering etc.) is also identified as a challenge.

Multiple issues associated with *land* have been identified by the survey respondents. One challenge that is mentioned multiple times is the current derelict condition of the site.

| Governance | Land | Finance | Design | Sustainability |
|---|---|---|---|---|
| **G1** Co-ordinated planning between administrations | **L1** Present derelict condition of the site | **F1** Economic/financing difficulties | **D1** Proportionate design of the site to fit the scale of the surroundings | **S1** Ensuring sustainability in exploitation economics<br>· Preferring alternative more income generating land use |
| **G2** Lack of speed in development ue to logistical complicacy and delays | **L2** Ownership of the land | **F2** Lack of resources for site development | **D2** Lack of knowledge and acceptance | **S2** Strict competition over land with other plausible land use<br>· Green land use vs other more economically viable alternative such as residence or parking |
| **G3** Ensuring maintenance (e.g. withstanding large number of visitor, damage with time, littering, etc.) | **L3** High density of the locality | | **D3** Achieving both functionality and aesthetics | **S3** Present soil contamination<br>· affecting the possibility of designing the green area for cultivation and residence |
| | **L4** Ongoing infrastructure projects around the site | | **D4** Stormwater management with regard to the correct dimensioning | |
| Proximity to a road and a railway (Hamnbanan and Lundbyleden) | | | | |
| **G4** Planning restriction on land use | **L5** Higher risk of accidents connected to transportation of dangerous goods | **F3** Potential policy conflict that can lead to large cost in future | **D5** Negatively affecting the possibility of designing a pleasant green area | **S4** Noise pollution |
| | **L6** Low connectivity to smaller roads and walkways | | | **S5** Air pollution |

**Fig 6. Identified challenges in realising UGS in the study site.**

Ongoing infrastructure projects around the site are also seen as a challenge, possibly because of the uncertainty they create in relation to the site's future. The site is in a densely populated neighbourhood which is also seen as a challenge since, after the completion of the Karlastaden project, the area will become even more built-up.

Three respondents identify *finance* as a challenge and one respondent further describes the lack of resources for developing a greenspace.

Several *design* challenges are recognised by the survey respondents. Ensuring that the realised UGS is both aesthetic and functional is considered challenging by some survey respondents. One respondent identifies a design issue specific to the site condition: the neighbouring construction of the seventy-three storey Karlatornet is, when completed, going to be the tallest tower in the Nordic region and so a proportionate design of the site to fit the scale of such a building is going to be a unique challenge for the site's designers. One respondent wonders how the shadow cast by this tall tower might impact the use of the site. Some design challenges are specific to certain green land use, such as correct dimensioning of bioswales for stormwater management.

Several respondents express their concern over ensuring *sustainability* in a market-driven urban economy, and three respondents further explain that strict competition over urban land might favour other, more immediately profitable, alternatives to greenspaces such as residential use or parking. Three respondents also identify the present soil contamination as potentially limiting the realisation of certain greenspaces e.g. allotments.

However, the most prominent issue for the study site is the road and railway that border the north of the site. Various challenges arise due to their presence adjacent to the site, with these issues spanning all categories and identified by ten respondents. Environmental concerns, such as noise and air pollution, are frequently mentioned by the respondents (*sustainability*). The proximity to busy roads and railway also means a higher risk of accidents (*land*). One respondent from the municipality explains that the risks associated with accidents regarding transportation of dangerous goods on the adjacent road puts the site into a high-risk zone.

The current detailed plan therefore restricts future uses of the site, such as using it for 'extended stay' (*governance*). This implies that future UGS should not be planned to allow people to stay there, but only to pass through, if no measures are implemented to lower the risk. Another challenge is the barrier effect, meaning low connectivity to smaller roads and walkways, making it difficult to connect the site to the neighbourhood (*land*). These complications can also lead to potential policy conflicts that could entail significant cost to resolve in future (*finance*). The presence of heavy traffic thus creates *design* challenges by negatively affecting the design of a pleasant green area.

## 3.4 Matching challenges with resources

**3.4.1 A developed method for matching challenges with resources.** The developed method consists of three steps:

1. A timeline for UGS development is established, consisting of four consecutive phases: (i) *planning* phase, (ii) *design* phase, (iii) *building* phase, and (iv) *use and maintenance* phase.

2. The identified challenges are regrouped according to the phases of the timeline.

3. The stakeholders, and the resources they can contribute, are placed in the timeline. Stakeholders' engagement in each phase of the timeline is based on the relevance of their resources to address the challenges associated with that phase.

The final output of this method is a timeline showing the stakeholders' involvement in each phase of an UGS development.

**3.4.2 Timing of resource mobilisation to cope with challenges at Polstjärnegatan.** The challenges identified in the study area are regrouped according to the different phases of the UGS development timeline (see Fig 7). They are referenced with their category and number, derived from the table in Fig 6, where the first governance challenge is coded as G1 and so on. The first two phases of the timeline, *planning* and *design*, include the highest number of challenges. Challenges with a broader scope need to be addressed in the planning phase, where the conceptualisation of the UGS takes place. Some of the challenges involving sustainability (S1, S2), governance (G1, G4), land (L1, L2), and finance (F1, F3) are placed in this phase. The design phase includes challenges that are more specific and relevant when a specific UGS has been decided on. All design challenges identified in the study area (D1–D5) as well as certain land (L3, L5, L6) and sustainability (S3–S5) challenges are grouped into this phase. The third phase of UGS development, the building phase, includes three challenges, one each from governance (G2), land (L4), and finance (F2). The final phase is the use and maintenance phase; there is only one challenge from the governance category (G3) included in this phase, about ensuring the maintenance of the realised UGS.

The stakeholders and the resources they can contribute with are included in the timeline visualisation, based on their ability to address the identified challenges of the various phases. As the landowner and the primary financier of the site development, the Real Estate Office is preferably engaged in all four phases of the timeline. The Parks and Nature Administration should also be involved throughout the timeline as the responsible authority of greenspaces in the city and be engaged in the realisation and maintenance of any UGS realised in the study area. The other municipal departments are likely to be engaged sporadically throughout the UGS development phases. The City Planning Authority and the Environmental Administration are preferably engaged in the first two phases when the conceptualisation and design of the UGS are being decided upon. In these phases, they can provide specific knowledge on municipal (and national) regulations, guidelines, and plans. The

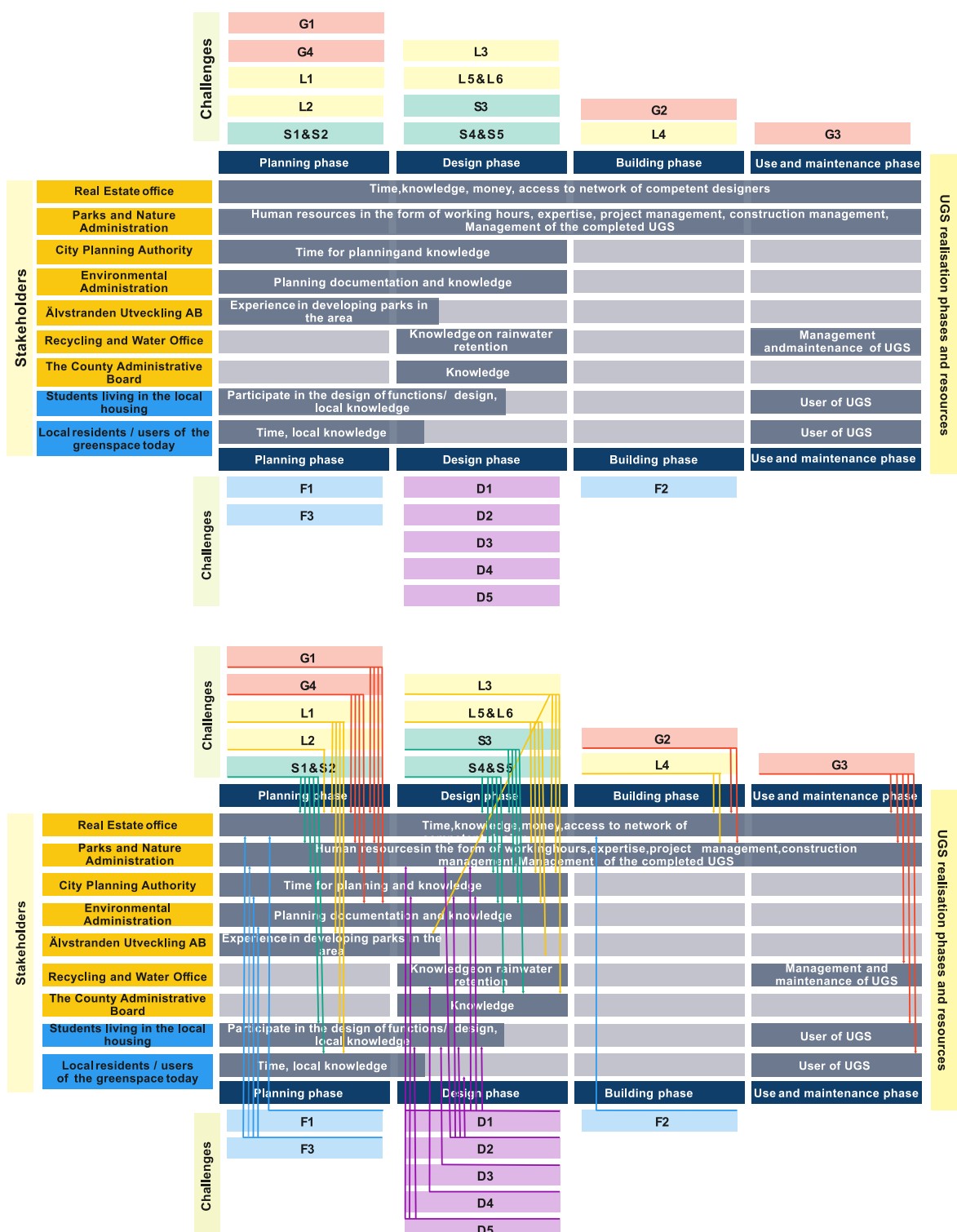

**Fig 7. Overlapping challenges with available resources; bottom figure shows all the connections between challenges with respective resources.**

municipal company Älvstranden Utveckling AB should also be engaged initially as they can provide relevant insights, based on their previous experience of developing parks in the neighbourhood. The involvement of Water and Waste Recycling Department is likely to depend on what type of UGS is decided upon. If the planned UGS is primarily designed to handle stormwater runoff (i.e. bioswale) or has features to carry out similar purposes, they need to be involved during the design and maintenance phase. Stakeholders representing regional government, the County Administration Board, are involved only during the design phase for specific inputs regarding national guidelines and regulations that need to be followed.

Local community stakeholders have expanded on a variety of resources they can bring forward in realising UGS in the study site. They are very well informed about local context and local needs, which could prove to be valuable inputs during the planning phase of the UGS development. If UGS with urban agriculture features are chosen to be developed on the site, local residents with experience in such activities can provide insights valuable also for the design phase. Ultimately, the local community represents the majority of the users of the realised UGS, and are essential in the final phase, use and maintenance.

## 4. Discussion, conclusions, and future work

This study presents a working process of how the selected and developed methods can be used as tools for understanding the various stakeholders' preferences and potential involvement in UGS development. The methods selected from the literature review were adapted to suit the aim of the research. Gigamapping, as used in this study, integrated more visual and graphical elements into the modified methods compared to the original methods. For example, the visual exploration of the Gigamap helped to realise the hidden potential in the collected data and the modified methods (Section 3.2.1) by mapping the interests and resources graphically. However, some remaining challenges using the modified Crosby method were not possible to overcome with Gigamapping. The definitions of the resources were limiting (time, knowledge, money) and, based on the responses where respondents expanded on the scope of the resources they could bring, it is clear that there is a need for better descriptions and categorisation of the resources needed. The suggested new method to match the identified challenges with available resources was developed to complement the methods for identifying resources and mapping challenges and to add an important layer to the analysis. The application of this method does not require additional data input. It helps to analyse whether the available resources may tackle the identified challenges.

The application of the methods at the study site Polstjärnegatan specifically led to the following insights:

- Stakeholders relevant for realising greenspaces at the site were effectively mapped in a Venn diagram to visualise stakeholder categories, stakeholder groups and stakeholders (Fig 3). The Venn diagram allows for visualising the interconnections and overlaps of different stakeholder roles. Here, the municipal/local government and local community were identified as the two most relevant stakeholder groups, with the Real Estate Office (the landowner of the site) tying together the two groups.

- The UGS most preferred by the stakeholders (Fig 4) is an urban park (selected by 26 out of 31 respondents) followed by bioswale (selected by 19 respondents) and community garden (selected by 12 respondents). From the comments it can be understood that the combination of several uses, especially integrated with urban park, is seen as positive, which may be an important input for the design.

- Stakeholders' resources (money, knowledge, time) were mapped to facilitate an understanding of how different stakeholders may contribute to realising the chosen USG (Fig 5). The Crosby categorisation of resources (time, knowledge, money) was limiting, but the respondents provided more in-depths reply in the commentary. At Polstjärnegatan, the Real Estate Office is the main source of money for the site's development, but other stakeholders can also bring in a multitude of other resources.

- The challenges associated with the development of the site into an UGS were identified and categorised (Fig 6). At Polstjärnegatan, the railway and highly trafficked motorway that border the site was identified as an aspect which creates multiple challenges associated with each one of the five categories: governance, land, finance, design and sustainability.

- Finally, a timeline over the different development phases and the needed mobilisation of resources among stakeholders to manage these challenges was visualised (Fig 7). The visualisation clearly shows the need for collaboration between stakeholders, both between departments in the municipality but also with external stakeholders to achieve an effective realisation of UGS.

The survey received quite a limited number of responses, although the responses given did provide a lot of information. The Recycling and Water Office was acknowledged as an important stakeholder by other respondents, especially for the bioswale option, but the office unfortunately did not respond to the survey. The Environmental Department at the municipality have different functions, and it is a bit uncertain whether e.g. the part that acts as controlling authority, is represented. They could also have given input regarding the view on any constraints a potential contamination situation would pose.

For the present study, the data collection method was primarily limited to the online questionnaire survey due to the ongoing pandemic. Therefore, it was at the time not possible to explore other alternative data collection methods. Other methods of data collection could potentially be better suited. Data collection for resource and challenge identification could also potentially be combined to become more efficient: stakeholders first identify resources and challenges and then work together to map challenges with resources. Targeted users of the working process are primarily practitioners in the field of urban planning, and potentially policy makers, for better understanding the challenges of incorporating UGS in cities and transforming underused lands to greenspaces. These methods can be used, for example, to engage citizens with decisions on public greenspaces in cities. Guidelines are already available for facilitating citizen participation on developing public spaces [54–56]. An example of this is provided by the pan-European GREEN SURGE project, which not only helped to create an inventory of the UGS in European cities [57], but also provided processes to link government-led planning with local communities and a framework for evaluating urban green infrastructures [58]. With a focus on the Nordic region, Molin et al. [59] developed a policy brief for citizen participation in UGS development. These studies suggest that contextual institutional challenges need to be considered for the methods to be useful tools for local governments. Kabisch [60] explained that, relating to Berlin in Germany, awareness of green benefits among different stakeholders is low and there are very few existing informal strategies that explicitly identify such ecosystem services. However, the respondents in our survey showed high awareness of the green benefits, even among non-professionals. Implementing the proposed methods on more sites would enable practitioners to map the awareness and acceptance of different greenspaces by the public at large. If the municipality or local government in question has proper regulations in place, or has enough autonomy and interest in UGS development, then the methods developed in this study can facilitate their needs.

In a review on the challenges associated with UGS planning in cities, Haaland & van den Bosch [13] confirms that densification processes such as infill development and consolidation can pose a threat to UGS. The site in this study is in a rapidly developing and densifying urban district and is part of a development project, Karlastaden, that consists of eight urban blocks of mixed commercial and residential development [49, 50]. Even though the site is designated to be designed as a greenspace, it can be considered as part of a larger urban densification project. With the speed of urbanisation in the area, the concern of the local stakeholders (Challenge S1 and S2 in Fig 6) that greenspaces in the area can come under more pressure to be repurposed for more economically beneficial (e.g. housing or commercial) land use. The dense Karlastaden comes with another challenge is that it consists of four high-rises and impact of such in planning the greenspace. Although unique to the local context, high-rise neighbourhoods exist in many cities across the world and the design challenges are studied for different contexts [61, 62]. There are now more subtler challenges, as explained by Colding et al. [9], that can conjugate and pose a greater threat in realising and maintaining UGS in cities. Challenges regarding ensuring maintenance (Challenge G3 in Fig 6) that stems from lack of financial resources (Challenge F1 and F2) can potentially lead to outsourcing the management to private sector. Such challenges as observed by the stakeholders at the study site are argued by Colding et al. [9] to result in gradual loss of access to greenspaces across many cities due to privatisation. This can negatively affect the urban inhabitants, specifically for citizens in Swedish cities, who largely associate their well-being with access to natural areas [63] and make frequent visits to different greenspaces located within the cities [64]. Even with almost 70% of land area covered with forest in Sweden [65], more than 50% of forest visits happen in the urban forests with most frequent visits occurring in the ones located within 1 km of the residents' home [64, 66]. At the study site, the respondents also emphasised on the importance of UGS by reasoning how stakeholders across all domains potentially benefit from them. The challenges identified for UGS realisation by the respondents as well as the resources they bring to tackle the challenges can help to support UGS realisation and maintenance in cities.

This study explicitly did not study the issues regarding contamination at the study site and the contamination status of the site was not shared with the respondents. Without an in-depth explanation of the reason (e.g. site history) and the risk posed by the contamination (e.g. by means of a risk assessment), there are possibilities of miscommunication that can trigger a negative stigma associated with contaminated sites. The contamination and associated remediation issue of the site was however, brought forward as a challenge by the survey respondents. The traditional, and still the most common way in Sweden, of dealing with contamination is to replace all contaminated soil with new clean soil [67, 68]. Such 'dig and dump' clean-up can be considered a linear 'take-use-dispose' approach to soil, where soil resource is treated to be unlimited and thus, disposable [21, 69]. But soil is a limited resource and due to its very slow regeneration it can be considered a non-renewable resource, a resource that at the same time is supporting the terrestrial nature [20]. A more circular approach would be to treat the soil in-situ with renewable nature-based resources. Gentle remediation options (GRO) are defined as a type of nature-based solutions that have the potential to manage risks to health and ecosystems while also improving the soil quality [70]. When transforming a brownfield into UGS, there are potential opportunities to apply GRO as a nature-based solution to deal with contamination [21]. The European Greenland project developed the Brownfield Opportunity Matrix (BOM) to link various services provided by soft reuse of brownfields to the potential measures that could be taken, as a way to engage and communicate among stakeholders [71]. Another development is made by Drenning et al. [72] where a generic GRO risk management framework is presented to support communication of the potential of different GRO strategies in managing ecological and human health risks posed by the contamination. Site-specific

application of the framework can support the identification of nature-based solutions that can be suitable for a specific site given specific UGS land uses, and Drenning et al. [72] demonstrates the framework in the context of the Polstjärnegatan site. If nature-based solutions are to be combined with UGS for brownfield regeneration, an important input will be the needs and preferences of stakeholders in order to be able to design such UGS site [73] This research paper focuses on exploring the perspective of stakeholders in terms of their preferences of a site's provision of various services. A further development of this work is to explore how possible nature-based solutions for managing the contamination at the site could be incorporated into the design of the preferred UGS.

The working process developed in this study for identifying and categorising stakeholders and their interests, challenges, and resources, can facilitate effective and realistic realisations of UGS in the context of regenerating urban brownfield sites. It demonstrates how multiple methods can be used in concert to map stakeholders, preferences, challenges, and resources for a particular site. For another site, other methods may be more suitable depending on the particular setting. Specifically, the Crosby method was found to be limiting for this site, although the survey did provide more in-depth information in the open-ended questions. The demonstration at the study site provided site-specific data but, the developed stakeholder categories and groups visualised in the Venn diagram, as well as the timeline of mobilisation of the resources of stakeholders to manage the challenges of realising an UGS, can be generalised to applications at other sites in Sweden and beyond.

## Supporting information

**S1 File. Final Gigamap.**
(PDF)

**S2 File. Sample questionnaire in English.**
(DOCX)

## Acknowledgments

The authors would like to thank the survey respondents, the three reviewers who with their constructive feedback have helped to improve the manuscript immensely.

## Author Contributions

**Conceptualization:** Shaswati Chowdhury, Jaan-Henrik Kain, Marco Adelfio, Yevheniya Volchko, Jenny Norrman.

**Data curation:** Shaswati Chowdhury.

**Formal analysis:** Shaswati Chowdhury.

**Funding acquisition:** Jaan-Henrik Kain, Marco Adelfio, Yevheniya Volchko, Jenny Norrman.

**Investigation:** Shaswati Chowdhury.

**Methodology:** Shaswati Chowdhury, Jaan-Henrik Kain, Marco Adelfio.

**Supervision:** Jenny Norrman.

**Validation:** Shaswati Chowdhury.

**Visualization:** Shaswati Chowdhury.

**Writing – original draft:** Shaswati Chowdhury.

**Writing – review & editing:** Shaswati Chowdhury, Jaan-Henrik Kain, Marco Adelfio, Yevheniya Volchko, Jenny Norrman.

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
