## [Decision Letter · Decision Letter 0]

22 Mar 2022

PONE-D-22-04756Transforming brownfields into urban greenspaces: Development and application of a multi-method approach for Stakeholder analysisPLOS ONE

Dear Dr. Chowdhury,

Thank you for submitting your manuscript to PLOS ONE. After careful consideration, we feel that it has merit but does not fully meet PLOS ONE’s publication criteria as it currently stands. Therefore, we invite you to submit a revised version of the manuscript that addresses the points raised during the review process.

While the reviewers felt that the subject of this manuscript was significant and of interest, there are a number of major weaknesses that will require an overhaul and restructuring of the manuscript. Please be sure to carefully address the criticisms and recommendations from the three reviewers, particularly reviewers 1 and 2. Make note of the following points from the reviewers:

The analysis is weak in terms of supporting the argument and conclusions of the paper.The ad-hoc selection of assessments does not support conclusions of a generalizable method.It is unclear that the methods chosen were the best suited for the study.It is unclear why this site was chosen for a conflict of interest study.The study aims to develop an approach to identify and categorize stakeholders and their interests, challenges, and resources but it is unclear how they are linked to the overall goal?The results section is a detailed description of the case study and the stakeholders, rather than a comparative evaluation of the different methods.The figures are overwhelming and use text that it is difficult to read.The manuscript is too long and needs a sharper focus.Sharpen focus on the comparative and rigorous assessment of different methods, or on the more questions of stakeholder preferences in the context of greenspace development from urban brownfields. Please submit your revised manuscript by May 06 2022 11:59PM. If you will need more time than this to complete your revisions, please reply to this message or contact the journal office at plosone@plos.org. Please include the following items when submitting your revised manuscript:A rebuttal letter that responds to each point raised by the academic editor and reviewer(s). You should upload this letter as a separate file labeled 'Response to Reviewers'.A marked-up copy of your manuscript that highlights changes made to the original version. You should upload this as a separate file labeled 'Revised Manuscript with Track Changes'.An unmarked version of your revised paper without tracked changes. You should upload this as a separate file labeled 'Manuscript'.

We look forward to receiving your revised manuscript.

Kind regards,

Theodore Raymond Muth

Academic Editor

PLOS ONE

Journal Requirements:

2. Please include in the Methods section of your manuscript text the following:

a) A brief statement regarding exemption from ethical review for this study under Swedish law, including the link to the relevant legislation you have provided in the Ethics Statement of the online submission information.

b) A brief statement regarding participant consent as noted in the Ethics Statement of the online submission information. Additionally, for the verbal consent obtained, please state how the consent was documented and witnessed.

[This work is supported by Formas (Grant number: 2017-00246).]

 [YES

This work is supported by Formas (Grant number: 2017-00246)]

[NO

The authors declare that they have no known competing financial interests or personal relationships that could have appeared to influence the work reported in this paper].

5. We note that Figure 3 in your submission contain map/satellite images which may be copyrighted. All PLOS content is published under the Creative Commons Attribution License (CC BY 4.0), which means that the manuscript, images, and Supporting Information files will be freely available online, and any third party is permitted to access, download, copy, distribute, and use these materials in any way, even commercially, with proper attribution. For these reasons, we cannot publish previously copyrighted maps or satellite images created using proprietary data, such as Google software (Google Maps, Street View, and Earth). For more information, see our copyright guidelines: http://journals.plos.org/plosone/s/licenses-and-copyright.

a) You may seek permission from the original copyright holder of Figure 3 to publish the content specifically under the CC BY 4.0 license.  

Reviewers' comments:

Reviewer's Responses to Questions

**Comments to the Author**

1. Is the manuscript technically sound, and do the data support the conclusions?

Reviewer #1: Partly

Reviewer #2: Partly

Reviewer #3: Yes

2. Has the statistical analysis been performed appropriately and rigorously? 

Reviewer #1: N/A

Reviewer #2: N/A

Reviewer #3: N/A

3. Have the authors made all data underlying the findings in their manuscript fully available?

Reviewer #1: Yes

Reviewer #2: Yes

Reviewer #3: Yes

4. Is the manuscript presented in an intelligible fashion and written in standard English?

Reviewer #1: Yes

Reviewer #2: Yes

Reviewer #3: Yes

5. Review Comments to the Author

Reviewer #1: Firstly, I want to thank the authors for their manuscript. This article addresses challenges posed by conflicts of interest among stakeholders regarding the conversion of brownfield to urban green spaces. To deal with these challenges, the authors propose to develop a multi-method stakeholder analysis. This involves reviewing available methods, compiling and modifying them as needed, and testing the results on a case study in Gothenburg. I found this topic relevant and interesting and the manuscript well-written.

However, I found the analysis to be weak in terms of supporting the argument and conclusions of the paper. In particular, I would direct attention to the following:

• While I find the idea of a scoping review to be completely valid, I found this to be less indicated by the framing of constructing a multi-method approach to brownfield stakeholder analysis. I would assume from such a framing that the authors were either (a) testing different methods or (b) applying pre-determined, standardized criteria to determine which methods are best suited to brownfield analysis generally. The somewhat ad-hoc selection presented here, while perfectly valid as the result of a scoping review to determine a method for this particular site / assessment does not, in my opinion, support conclusions of a generalizable method.

• Moreover, it is unclear that the methods chosen were the best suited. First, the Crosby method had to be substantially modified to fit the research design and constraints, raising questions as to its suitability when compared to other methods. Second, the survey itself yielded a relatively low sample size, with a low diversity of stakeholders. This in turn does not, in my estimation, support claims for the validity of these methods as best suited for the site, nor generalizable to brownfields broadly.

• Regarding the site, it is unclear why this site was chosen for a conflict of interest study. Given the input of the stakeholders in the survey, there do not appear to be deep conflicts regarding the future of the site. In the background, we are not introduced to any controversies or disagreements about the redevelopment of the site. Likewise, the manuscript is very focused on conflicts regarding what kind of UGS brownfields should be redeveloped into, which elides very common conflicts that arise around whether these sites should become UGS at all. Development pressures are mentioned in ln. 75, but we very quickly move away from the possibility of conflicts regarding UGS and say, housing stock. This relates further to a lack of attention to the political and economic context in which stakeholders are operating, as well as power within these political and economic contexts as a resource that some stakeholders have and others lack.

• Finally, there is a lack of attention to ecological stakeholders—for example, specialists in remediation, climate change adaptation, biodiversity conservation, landscape architecture, to say nothing of more-than-human stakeholders such as plants or wildlife. While presumably many of these stakeholders could come in via the categories presented in the paper, they did not, nor was their absence acknowledged or addressed, which is to my mind a critical gap. Given that the attention is to brownfields, which are often contaminated and in need of remediation, this seems very important to address.

Given these issues, I would recommend a rather major reframing of the paper. The central contribution appears to be the use of multiple stakeholder analysis instruments to map resources onto challenges. This is of interest and a contribution. I would suggest the following changes:

• Rather than present the analysis as the development of a new, mixed-method approach, I would suggest framing as a model for how multiple instruments / tools can be used in concert to maps resources and challenges among stakeholders. As such, I would not suggest the general applicability of this particular set of instruments, but rather the overall process engaged here as a model process that others could use with reference to their particular sites.

• I would provide more context on conflicts regarding the site, and if there are none reframe as not about conflicts of interests, but as a tool for how multiple stakeholders can best map resources to challenges.

• Address the weaknesses in terms of low diversity of stakeholders and what is lost from the analysis as a result.

Two further overall, though more minor, issues:

• Characterizing the study as mixed-methods is to me a little misleading. The method—stakeholder analysis via survey—is unified. What is mixed is the different instruments for stakeholder analysis being used. I would assume a mixed-method study to use different qualitative and/or quantitative instruments or analyses (e.g. ranked choice surveys and interviews).

• The focus in Gigamapping was confusing, as it was not one of the methods (or instruments) being evaluated and modified, but rather an internal tool for supporting and organizing the analytic work. As such, I would expect it to be briefly mentioned in the methods, but not named in the Abstract and Intro, no given a full section in the Methods.

More specific edits:

• Ln. 61-65: addresses densification as a pressure on UGS, but what of the drivers of densification, which are often motivated by sustainability goals? In this calls for more density and calls for more UGS are emerging from the same overall goals (sustainability). I think more nuance in the discussion of density and UGS is needed here.

• Ln. 105-106: isn’t it also, especially in the US, a function of decreased public resources, neoliberalization of the state?

• Ln. 179: you have a hanging sentence here

• Ln. 276, 293: can you be more specific about the contamination. Is it point source (as the metal and PCB from melting operations would suggest) or more broad (as the coal yard and sludge aspects would suggest)? In general, more attention to the specificities of contamination in brownfields and the significant role they play in shaping any redevelopment is warranted.

• Ln. 503, section 3.3.1 generally: what about the role of justice or equity in the process of planning and design?

Reviewer #2: The article “Transforming brownfields into urban greenspaces: Development and application of a multi-method approach for Stakeholder analysis” provides a massive effort in assembling information on stakeholders involved in the development of urban greenspace in a case study area in Gothenburg. The authors compare and apply multiple methods drawing on existing literature, an online questionnaire and expert-based system-mapping approaches. The combined effort of these different methods and outcomes is visualized in a Gigamap, a large and detailed visual collection of the study elements on a Miro board, illustrating the multiple interlinkages and connections of the study system.

While the integration of multiple stakeholders in greenspace development is certainly a complex task, the study left me a bit confused why the full range of methods and results presented here is needed and deemed to be helpful to solve this task. The study clearly states the goal to develop an approach to identify and categorize stakeholders and their interests, challenges, and resources with the aim to develop urban greenspaces from regenerating urban brownfields. The five more specific objectives are all related to the exploration of different methods, and it is unclear how they are linked to the overall goal and if the whole study is more about an evaluation of those methods or about the subject of greenspace development. Moreover, the results section is merely a detailed description of the case study and the stakeholders, rather than a comparative evaluation or triangulation of the different methods. Finally, the provided Figures are a bit overwhelming, as they accumulate a lot of information, with text so small that it is hardly legible.

Regarding the gigamapping approach, from the literature I understand this mostly as a tool for stakeholder engagement, where multiple people from various backgrounds work on one big picture of the system they operate in. The outcome of this is typically quite complex and convoluted and it requires scientists, planners or facilitators to synthesize the information and highlight what are the important streams of thought. Yet, to my understanding, this was not the approach that has been followed in this paper. Instead, the gigamap presented in the SI seems to be the product of the authors and if that is the case, I wonder what the justification is for choosing to present such a width of information rather than providing clarity. The tool used to produce the map in this case was Miro, an online whiteboard, and it seems possible to invite stakeholders to engage in an online discussion to produce such a Gigamap together even without meeting in person. I may have missed it, but I am not clear why this has not been done.

The manuscript is well-written language-wise, but it is extremely long and detailed, and it is hard to keep focus of what has been done for what purpose. The results section draws heavily on the results from the 31 online questionnaires, but a critical assessment is missing, how far this data is representative for what is going on in the case study or for what other purpose this data can be used.

Most Figures are extracts from the Gigamap in SI. It would help if the Figures for publication would be simplified and targeted towards the most important information. Figures 2, 7, and 8 are not legible in the current resolution and it is questionable if the shown amount of information will fit into a final figure. As with any map or visual representation, there is always a trade-off between depth and clarity. It is not a result in itself to represent complexity by a high number of linked items and information. This may be fine for the SI, but the manuscript figures need to summarize the most important results in a meaningful way and highlight which is the important information to support the statements in the text.

Overall, this work may well contain aspects that are worth publishing, but it would require major efforts in breaking down the text and the visual materials to follow a more coherent line of argument. This would ideally result in an article length that is more digestible, focusing either on the comparative (and rigorous) assessment of different methods, or on the more practical questions of stakeholder preferences in the context of greenspace development from urban brownfields.

Specific comments:

Title: The title implies that brownfields can be transformed into urban greenspaces and it seems the article takes this as a given. Yet, one may question why brownfields are not considered to already be greenspaces. Unfortunately, the manuscript does not cover this question, except a short mention of the need for decontamination of former industrial sites in the introduction.

L32: Unclear what is meant with the “realisation of a greenspace”. Does this mean to free up space in the competitive urban environment or does it mean to convert brownfields to parks? The latter would depend on the definition of what is considered “greenspace”.

L34(ff): What do you mean with “a modified method”? Which method and what kind of modification?

L36: Is "the site" the case study introduced in the next sentence or is it meant more generally?

L38: We are approaching the end of the abstract and this is still methods description. I would suggest to structure the abstract more or less equally into introduction, methods, results and discussion/conclusions.

L41: Unclear why this sentence starts with “however”, is there a contradiction with the previous sentence?

L84: Also brownfields already provide some important ecosystem services. Would be good to be more specific here, what kind of improvements derived from greenspaces are considered in the study.

L89-90: I don't understand why this sentence is needed. In the next paragraph you state that identifying and categorizing stakeholders is the main goal of this paper?

L113: So, is the methods development the purpose of the paper or is the purpose the identification and categorization of stakeholders in a case study? This difference is important, as the method should follow the purpose and not the other way around.

L127: Better to refer to a peer-reviewed reference for the Gigamapping approach

L129: Should “thus” be replaced by “then”?

L131: Maybe state already what kind of method was new

L137-139: I think here it will be enough to just report what has been done for what reason rather than listing all methodological options that were considered

L145: What is a scoping review and how is it carried out?

L155: Looking at the methods selection part in the SI (as Fig 2 is not legible) I do not understand why there is so much consideration of different methods if the Gigamapping tool seems to already have been decided on.

L171: Who is the research team and how is their expertise suitable or representative for the overall process that you are trying to analyze?

L176: It may not be enough to just mention the challenge of simplifying the gigamap, rather there should be some explanation how you dealt with this challenge.

L182: I agree that the gigamap is complex, but how does that make it useful to help solve the problem or reach the objectives of the study?

L266: There should be some consideration how far the number of respondents allows answering the questions. I do not see an issue with following a more qualitative approach and using the surveys as a proof-of-concept. Yet they may not be sufficient to quantitatively compare different groups of stakeholders.

L272: What about the remaining participants? Given the lack of balance between sampled groups, a comparison between stakeholder groups does not seem possible. Maybe best to lump all responses together as an indicator for general public interest?

L304: Do you mean "not all stakeholders could be categorized"?

L344: Fig. 4 looks nice but it does not look like a method, rather like a result. Also, it remains unclear from the methods, how this diagram has been developed and why you think it is conclusive

L321-382: This whole section is a very detailed description of the stakeholders in the case area and it is unclear, how this is an outcome of the methodology or supports the use of the method.

L399: What are medium answers?

L414: If there is no trade-off or cost involved in the decision, of course everyone would opt for a new park if asked.

L418: This already sounds like part of the discussion

L426: The painted pictures in Fig. 5 are very nice. If you have the rights to use them it would be good to make them much bigger in the Fig. to show some intuitive images to the reader

L427-469: This section is a description of different possible types of UGS in the study area, which is again not a result of the methodology applied.

L505-506: what is this categorization based on?

L607: Fig. 8 is almost impossible to read and understand

L650-666: This whole paragraph is on methods selection and not sure how far it fits in the discussion (given the current results section)

Reviewer #3: This innovative study aims to help facilitate the redevelopment of urban brownfields into urban green spaces (UGS) in urban metro areas by using literature review, mapping, and data analysis to 1) identify and characterize stakeholders; 2) map stakeholder interests and resources; 3) map challenges to brownfield redevelopment; and 4) match challenges with resources. The study draws case-specific conclusions about the effectiveness of each aforementioned method in n Polstjärnegatan, Gothenburg (SE). This paper has great potential to give important insights to brownfield redevelopment and greenspace development efforts. Below I outline some suggestions for consideration to improve the quality and impact of the manuscript.

Introduction

• I suggest updating your references on the influence or urban greenspace on mental and physical health. Your most recent reference (in line 49) is 2010. Below are some good empirical articles.

• You briefly mention that brownfields are potentially contaminated.

• In line 98, do you mean members of the public or the community? “Members of the society” can have a very broad meaning. I consider everyone, everything, and every organization as a part of society. Be more specific here.

Methodology

• It is unclear in the description of stakeholder analysis whether the UGS that respondents selected as ‘interest’ where actual existing UGS or possible locations for UGS/former brownfields. Authors should be more explicit in describing this question/process in lines 210-220.

• Authors should further describe the reference group that was given the survey initially. See lines 252-254 “The online data collection process started with the members of a reference group to the research project…… who were most municipality representatives.”

• Did the authors collect any other demographics from the sample (e.g., race/ethnicity, age, # of years in residence, gender, education)? Factors such as these might be important for understanding responses, interests, and challenges.

Results:

• Here in the results the reader finally gets a description of UGS interests. These are the types of UGS that the stakeholder prefers. A brief description of the type options should be placed in the methodology section. (See first bullet point under methodology)

Discussion:

• I notice that there is not much discussion of contamination of the brownfield and the health impacts of its redevelopment into UGS. Authors should at least discuss this and perhaps why this didn’t present itself as much in the “challenges” for stakeholders. Are stakeholders aware of potential exposure and risks to health?

• Perhaps racial, gender, and socioeconomic dynamics do not play as large a role in the local contexts (Sweden) as it does in U.S. contexts. As the authors note that their findings can and should be generalizable to broader contexts involving urban brownfield redevelopment into greenspace, it might be worthwhile to at least discuss how different social contexts could shape findings.

6. PLOS authors have the option to publish the peer review history of their article (what does this mean?). If published, this will include your full peer review and any attached files.

Reviewer #1: No

Reviewer #2: No

Reviewer #3: No

---

## [Author Response · Author response to Decision Letter 0]

2 Jun 2022

'A separate file is attached titled 'response to review comments' which should be referred instead of this. The comments are replied using tables and this box does not support that.'

PONE-D-22-04756

Transforming brownfields into urban greenspaces: A working process for stakeholder analysis (revised from: Transforming brownfields into urban greenspaces: Development and application of a multi-method approach for Stakeholder analysis)

PLOS-ONE

Dear Editor,

We have revised the article PONE-D-22-04756 in accordance with the changes suggested by the editor and the reviewers. We have explicitly addressed the specific comments of the reviewers, see details in Table 1. The comments were constructive and helped to improve the manuscript. 

Best regards,

The authors

Editor’s summary points

Table 1. Editor’s summary points and responses to how they are addressed. 

Comment How it is addressed

The analysis is weak in terms of supporting the argument and conclusions of the paper.

The ad-hoc selection of assessments does not support conclusions of a generalizable method.

It is unclear that the methods chosen were the best suited for the study.

The study aims to develop an approach to identify and categorize stakeholders and their interests, challenges, and resources but it is unclear how they are linked to the overall goal?

The results section is a detailed description of the case study and the stakeholders, rather than a comparative evaluation of the different methods.

Sharpen focus on the comparative and rigorous assessment of different methods, or on the more questions of stakeholder preferences in the context of greenspace development from urban brownfields. The overall aim and objective of the paper has been reformulated. The previous version stated: ‘The overall aim of the study was to develop a multi-method approach for identifying and categorising stakeholders and their interests, challenges, and resources, and thereby facilitate effective and realistic realisations of UGS in the context of regenerating urban brownfields. Five specific research objectives were formulated to address the overall aim: 

1) Identify and modify a suitable method for identifying and categorising relevant stakeholders for realising UGS on brownfields in cities 

2) Identify and modify a suitable method for mapping the interests and resources of such stakeholders

3) Identify and modify a suitable method for mapping the various challenges associated with realising UGS on brownfield sites in cities 

4) Develop a new method for matching the identified challenges with the stakeholder interests and resources available to realise UGS on brownfields in cities

5) Demonstrate the aforementioned methods by applying them to a case study and presenting the results.’

The revision of the aim and objectives is done in accordance with the suggestion by reviewer: 

‘Rather than present the analysis as the development of a new, mixed-method approach, I would suggest framing as a model for how multiple instruments / tools can be used in concert to maps resources and challenges among stakeholders. As such, I would not suggest the general applicability of this particular set of instruments, but rather the overall process engaged here as a model process that others could use with reference to their particular sites.’

The revised version states (L 124 – 133 of the revised manuscript): ‘The overall aim of this study was to support effective and realistic realisations of UGS in the context of urban brownfields’ regeneration and stakeholder engagement. To achieve this overall aim, a working process was developed to:

1. integrate various methods relevant for UGS realisation, including methods for a) identification and categorising of relevant stakeholders, b) mapping their interests and resources, c) identifying the various challenges associated with realising UGS on urban brownfields, and d) matching those challenges with the mapped resources over the timeline of UGS development; and

2. apply methods within this working process to assess its relevance and identify potential shortcomings.’

To support the revised aims and objectives, the text of the manuscript has also been revised. The key points are summarised below:

1. The methodology has been simplified and focus is now on developing a working process of different methods rather selecting suitable methods from literature. The exploration of literature for selecting suitable methods and modifying them has been removed (along with 2 of the previous manuscript). It is still part of the overall research work and is presented in the supplementary material (i.e. Gigamap). 

2. The result is also simplified and now presents only the output of applied or developed method on the case study.

3. The discussion is revised to reflect on the result, limitation of the study and the future possibilities. The points highlighted by the reviewers (e.g. too much focus on the selected method suitability, no mention of the ecological stakeholders and remediation) have been taken into consideration when restricting and re writing the discussion section.

It is unclear why this site was chosen for a conflict-of-interest study. The focus of the paper has been shifted from selection of methods and conflict of interest (see the row above) to exploring the stakeholders’ potential to contribute with resources to meet challenges and developing a working process to realise UGS on brownfields. The case Polstjärnegatan has been suggested originally by the municipality of Göteborg and has been used as case study in two previous papers and a Licentiate thesis (Chowdhury, 2020; Chowdhury et al., 2020; Drenning et al., 2022). 

The figures are overwhelming and use text that it is difficult to read. The figure 1 and 2 from the previous manuscript has been removed in the revised one. Figure 6 (previously Figure 8) which was pointed out for being difficult to read has been reworked and should be readable in the current version, 

The manuscript is too long and needs a sharper focus. The revised manuscript has been shortened 1045 words compared to the previous 9842 words, and stands at 8797 words without abstract and references. 

Reviewers’ comments

Color coding of how the points raised has been addressed:

Completely, Partially, Denied

Reviewer 1

General comment by reviewer: Firstly, I want to thank the authors for their manuscript. This article addresses challenges posed by conflicts of interest among stakeholders regarding the conversion of brownfield to urban green spaces. To deal with these challenges, the authors propose to develop a multi-method stakeholder analysis. This involves reviewing available methods, compiling and modifying them as needed, and testing the results on a case study in Gothenburg. I found this topic relevant and interesting and the manuscript well-written.

Table 2. Measures and responses to reviewer 1’s comments. The reviewer’s reference to lines in the manuscript are to the original manuscript. 

 Comments How it is addressed

Major issues While I find the idea of a scoping review to be completely valid, I found this to be less indicated by the framing of constructing a multi-method approach to brownfield stakeholder analysis. I would assume from such a framing that the authors were either (a) testing different methods or (b) applying pre-determined, standardized criteria to determine which methods are best suited to brownfield analysis generally. The somewhat ad-hoc selection presented here, while perfectly valid as the result of a scoping review to determine a method for this particular site / assessment does not, in my opinion, support conclusions of a generalizable method. In line with the reviewers suggestion we have slightly changed the framing of the paper. The selection of methods is indeed based on a scoping review, and the aim of the study has been revised. The revised aim and objectives states (L 124 – 133 in the revised manuscript): 

‘The overall aim of this study was to support effective and realistic realisations of UGS in the context of urban brownfields’ regeneration and stakeholder engagement. To achieve this overall aim, a working process was developed to:

1. integrate various methods relevant for UGS realisation, including methods for a) identification and categorising of relevant stakeholders, b) mapping their interests and resources, c) identifying the various challenges associated with realising UGS on urban brownfields, and d) matching those challenges with the mapped resources over the timeline of UGS development; and

2. apply methods within this working process to assess its relevance and identify potential shortcomings.’

 Moreover, it is unclear that the methods chosen were the best suited. First, the Crosby method had to be substantially modified to fit the research design and constraints, raising questions as to its suitability when compared to other methods. Second, the survey itself yielded a relatively low sample size, with a low diversity of stakeholders. This in turn does not, in my estimation, support claims for the validity of these methods as best suited for the site, nor generalizable to brownfields broadly. The reviewer has a good point and we therefore do not make the point that the selected methods are the best for all cases, but were suitable for this case. The Crosby method was indeed problematic and this is thoroughly discussed. 

 Regarding the site, it is unclear why this site was chosen for a conflict of interest study. Given the input of the stakeholders in the survey, there do not appear to be deep conflicts regarding the future of the site. In the background, we are not introduced to any controversies or disagreements about the redevelopment of the site. Likewise, the manuscript is very focused on conflicts regarding what kind of UGS brownfields should be redeveloped into, which elides very common conflicts that arise around whether these sites should become UGS at all. Development pressures are mentioned in ln. 75, but we very quickly move away from the possibility of conflicts regarding UGS and say, housing stock. This relates further to a lack of attention to the political and economic context in which stakeholders are operating, as well as power within these political and economic contexts as a resource that some stakeholders have and others lack. The focus is also shifted from ‘conflicts of interest’; this phrase is avoided and rather, the stress is put on the importance of stakeholders’ interests, resources, and collaboration. 

 Finally, there is a lack of attention to ecological stakeholders—for example, specialists in remediation, climate change adaptation, biodiversity conservation, landscape architecture, to say nothing of more-than-human stakeholders such as plants or wildlife. While presumably many of these stakeholders could come in via the categories presented in the paper, they did not, nor was their absence acknowledged or addressed, which is to my mind a critical gap. Given that the attention is to brownfields, which are often contaminated and in need of remediation, this seems very important to address. The discussion chapter currently includes a discussion on ecological stakeholders such as, remediation experts, comparing the results with contemporary research and what can be done in future (L 627 – 656 in the revised manuscript):

This study explicitly did not study the issues regarding contamination at the study site and the contamination status of the site was not shared with the respondents. Without an in-depth explanation of the reason (e.g. site history) and the risk posed by the contamination (e.g. by means of a risk assessment), there are possibilities of miscommunication that can trigger a negative stigma associated with contaminated sites. The contamination and associated remediation issue of the site was however, brought forward as a challenge by the survey respondents. The traditional, and still the most common way in Sweden, of dealing with contamination is to replace all contaminated soil with new clean soil (Edebalk, 2013; SGI, 2018). Such ‘dig and dump’ clean-up can be considered a linear ‘take-use-dispose’ approach to soil, where soil resource is treated to be unlimited and thus, disposable (Chowdhury, 2020a; Chowdhury et al., 2020). But soil is a limited resource and due to its very slow regeneration it can be considered a non-renewable resource, a resource that at the same time is supporting the terrestrial nature (Breure et al., 2018). A more circular approach would be to treat the soil in-situ with renewable nature-based resources. Gentle remediation options (GRO) are defined as a type of nature-based solutions that have the potential to manage risks to health and ecosystems while also improving the soil quality (Cundy et al., 2015). When transforming a brownfield into UGS, there are potential opportunities to apply GRO as a nature-based solution to deal with contamination (Chowdhury et al., 2020). The European Greenland project developed the Brownfield Opportunity Matrix (BOM) to link various services provided by soft reuse of brownfields to the potential measures that could be taken, as a way to engage and communicate among stakeholders (Beumer et al., 2014). Another development is made by Drenning et al. (2022) where a generic GRO risk management framework is presented to support communication of the potential of different GRO strategies in managing ecological and human health risks posed by the contamination. Site-specific application of the framework can support the identification of nature-based solutions that can be suitable for a specific site given specific UGS land uses, and Drenning et al. (2022) demonstrates the framework in the context of the Polstjärnegatan site. If nature-based solutions are to be combined with UGS for brownfield regeneration, an important input will be the needs and preferences of stakeholders in order to be able to design such UGS site (Cundy et al., 2016). This research paper focuses on exploring the perspective of stakeholders in terms of their preferences of a site’s provision of various services. A further development of this work is to explore how possible nature-based solutions for managing the contamination at the site could be incorporated into the design of the preferred UGS.

Suggestions • Rather than present the analysis as the development of a new, mixed-method approach, I would suggest framing as a model for how multiple instruments / tools can be used in concert to maps resources and challenges among stakeholders. As such, I would not suggest the general applicability of this particular set of instruments, but rather the overall process engaged here as a model process that others could use with reference to their particular sites. Agree, the paper is changed according to the reviewer’s suggestion. More detailed on Editor’s summary point table (Row 1, column 2). 

 • I would provide more context on conflicts regarding the site, and if there are none reframe as not about conflicts of interests, but as a tool for how multiple stakeholders can best map resources to challenges. To meet this comment, we have put less focus on conflicts as this indeed was not an issue of particular concern at this site (it will be a green site, but could of course have different green land uses), and more focus on how to map resources. 

 • Address the weaknesses in terms of low diversity of stakeholders and what is lost from the analysis as a result Agree. These lines are added in the discussion (L 597 – 602)

The survey received quite a limited number of responses, although the responses given did provide a lot of information. The Recycling and Water Office was acknowledged as an important stakeholder by other respondents, especially for the bioswale option, but the office unfortunately did not respond to the survey. The Environmental Department at the municipality have different functions, and it is a bit uncertain whether e.g. the part that acts as controlling authority, is represented. They could also have given input regarding the view on any constraints a potential contamination situation would pose. 

Minor issues Characterizing the study as mixed-methods is to me a little misleading. The method—stakeholder analysis via survey—is unified. What is mixed is the different instruments for stakeholder analysis being used. I would assume a mixed-method study to use different qualitative and/or quantitative instruments or analyses (e.g. ranked choice surveys and interviews) The title is now reworked reflecting the changes in the manuscript (see the previous row and the table of the editor’s comments) and now states: ‘Transforming brownfields into urban greenspaces: A working process for stakeholder analysis’.

 The focus in Gigamapping was confusing, as it was not one of the methods (or instruments) being evaluated and modified, but rather an internal tool for supporting and organizing the analytic work. As such, I would expect it to be briefly mentioned in the methods, but not named in the Abstract and Intro, no given a full section in the Methods. Focus from Gigamapping is greatly reduced in the revised manuscript. As suggested, 

• Gigamapping is not mentioned in the abstract or as a keyword

• Rather than a separate section on the method section, it is now just described in the Section 2: Methodology 

• The discussion on the gigamap in the result section has also been limited. 

However, the use of the Gigamapping was indeed important for the result of the paper. 

Specific issues Ln. 61-65: addresses densification as a pressure on UGS, but what of the drivers of densification, which are often motivated by sustainability goals? In these calls for more density and calls for more UGS are emerging from the same overall goals (sustainability). I think more nuance in the discussion of density and UGS is needed here. Thank you for bringing this perspective up. This line is added (L 66 – 68 in the revised manuscript):

‘The compact city is the by-product of the sustainable planning approach for urban development that has pushed for more densified cities to tackle urban sprawl (Haaland & van den Bosch, 2015).’

 Ln. 105-106: isn’t it also, especially in the US, a function of decreased public resources, neoliberalization of the state? Thank you for the interesting perspective. The following line is added (L 116 – 119 in the revised manuscript): 

‘Particularly for the US, neo-liberal policy making and practice have reduced public sectors’ engagement in the generation of green spaces contributing to their privatisation and commodification (Harvey, 2006; Roy, 2011).’

 Ln. 179: you have a hanging sentence here It is continuation of the previous line, the capital G in Gigamapping spelling may have caused the confusion. 

 Ln. 276, 293: can you be more specific about the contamination. Is it point source (as the metal and PCB from melting operations would suggest) or more broad (as the coal yard and sludge aspects would suggest)? In general, more attention to the specificities of contamination in brownfields and the significant role they play in shaping any redevelopment is warranted. Since this study just touches upon the contamination issue and it was not the information shared in detail in the stakeholder survey, we tried to minimise the information on the manuscript. I will refer to our previous work where the contamination issues have been discussed in detail. This line is added (L 262 – 263 in the revised manuscript): ‘For detailed information regarding the contamination situation at the Polstjärnegatan site, see Chowdhury (2020, pp. 41–43).’

 Ln. 503, section 3.3.1 generally: what about the role of justice or equity in the process of planning and design? Thank you for identifying this issue. As it is still the results section, it is difficult to put int perspective something the survey participants themselves did not bring forward. We have tried to incorporate it in the discussion but could not find a scope to do so without risking the manuscript to become again, too long. We have brought forward the major issue pointed out i.e the lack of ecological stakeholder and added a paragraph on that in the discussion (row 4, column 3)

Reviewer 2

The article “Transforming brownfields into urban greenspaces: Development and application of a multi-method approach for Stakeholder analysis” provides a massive effort in assembling information on stakeholders involved in the development of urban greenspace in a case study area in Gothenburg. The authors compare and apply multiple methods drawing on existing literature, an online questionnaire and expert-based system-mapping approaches. The combined effort of these different methods and outcomes is visualized in a Gigamap, a large and detailed visual collection of the study elements on a Miro board, illustrating the multiple interlinkages and connections of the study system.

Overall, this work may well contain aspects that are worth publishing, but it would require major efforts in breaking down the text and the visual materials to follow a more coherent line of argument. This would ideally result in an article length that is more digestible, focusing either on the comparative (and rigorous) assessment of different methods, or on the more practical questions of stakeholder preferences in the context of greenspace development from urban brownfields.

Table 3. Measures and responses to reviewer 2’s comments. The reviewer’s reference to lines in the manuscript are to the original manuscript. 

 Comments How it is addressed (some comments are descriptive and does not require action, so those were marked ‘partially’ as well)

Major issues While the integration of multiple stakeholders in greenspace development is certainly a complex task, the study left me a bit confused why the full range of methods and results presented here is needed and deemed to be helpful to solve this task. The study clearly states the goal to develop an approach to identify and categorize stakeholders and their interests, challenges, and resources with the aim to develop urban greenspaces from regenerating urban brownfields. The five more specific objectives are all related to the exploration of different methods, and it is unclear how they are linked to the overall goal and if the whole study is more about an evaluation of those methods or about the subject of greenspace development. Moreover, the results section is merely a detailed description of the case study and the stakeholders, rather than a comparative evaluation or triangulation of the different methods. Finally, the provided Figures are a bit overwhelming, as they accumulate a lot of information, with text so small that it is hardly legible. The manuscript has been revised to address the shortcomings addressed here and now presents a restructured and reduced text with a simpler structure. The revised aim and objectives states (L 124 – 133 in the revised manuscript): ‘The overall aim of this study was to support effective and realistic realisations of UGS in the context of urban brownfields’ regeneration and stakeholder engagement. To achieve this overall aim, a working process was developed to:

1. integrate various methods relevant for UGS realisation, including methods for a) identification and categorising of relevant stakeholders, b) mapping their interests and resources, c) identifying the various challenges associated with realising UGS on urban brownfields, and d) matching those challenges with the mapped resources over the timeline of UGS development; and

2. apply methods within this working process to assess its relevance and identify potential shortcomings.’

More details on the editor’s comments

The images have been revised and reworked for their visibility. One image has been removed altogether.

 Regarding the gigamapping approach, from the literature I understand this mostly as a tool for stakeholder engagement, where multiple people from various backgrounds work on one big picture of the system they operate in. The outcome of this is typically quite complex and convoluted and it requires scientists, planners or facilitators to synthesize the information and highlight what are the important streams of thought. Yet, to my understanding, this was not the approach that has been followed in this paper. Instead, the gigamap presented in the SI seems to be the product of the authors and if that is the case, I wonder what the justification is for choosing to present such a width of information rather than providing clarity. The tool used to produce the map in this case was Miro, an online whiteboard, and it seems possible to invite stakeholders to engage in an online discussion to produce such a Gigamap together even without meeting in person. I may have missed it, but I am not clear why this has not been done. Focus from Gigamapping is greatly reduced in the revised manuscript. As suggested, 

• Gigamapping is not mentioned in the abstract or as a keyword

• Rather than a separate section on the method section, it is now briefly discussed in the Section 2: Methodology 

The discussion on the gigamap in the result section has also been limited. 

 The manuscript is well-written language-wise, but it is extremely long and detailed, and it is hard to keep focus of what has been done for what purpose. The results section draws heavily on the results from the 31 online questionnaires, but a critical assessment is missing, how far this data is representative for what is going on in the case study or for what other purpose this data can be used. Indeed, we have been able to reduce the amount of text. The revised manuscript has been shortened 1045 words compared to the previous 9842 words, and stands at 8797 words without abstract and references. 

To address what is missing or the limitation, we have added these lines in the discussion (L 597 – 602): ‘The survey received quite a limited number of responses, although the responses given did provide a lot of information. The Recycling and Water Office was acknowledged as an important stakeholder by other respondents, especially for the bioswale option, but the office unfortunately did not respond to the survey. The Environmental Department at the municipality have different functions, and it is a bit uncertain whether e.g. the part that acts as controlling authority, is represented. They could also have given input regarding the view on any constraints a potential contamination situation would pose.’

 Most Figures are extracts from the Gigamap in SI. It would help if the Figures for publication would be simplified and targeted towards the most important information. Figures 2, 7, and 8 are not legible in the current resolution and it is questionable if the shown amount of information will fit into a final figure. As with any map or visual representation, there is always a trade-off between depth and clarity. It is not a result in itself to represent complexity by a high number of linked items and information. This may be fine for the SI, but the manuscript figures need to summarize the most important results in a meaningful way and highlight which is the important information to support the statements in the text. Figure 2 of the previous manuscript has been removed.

Figure 7 and 8 (Figure 6 and 7 in the revised manuscript) has both been reworked for their visibility. 

The gigamap provided as a supplementary material is already a synthesized version of the work that took place on Miro. The figures in question are output of several synthesis diagrams and flow charts. However, as suggested, the issue with them being clear still stands and the images are reworked for their resolution, visibility, and readability. 

Specific issues Title: The title implies that brownfields can be transformed into urban greenspaces and it seems the article takes this as a given. Yet, one may question why brownfields are not considered to already be greenspaces. Unfortunately, the manuscript does not cover this question, except a short mention of the need for decontamination of former industrial sites in the introduction. The UGS categorisation we are using (Urban GI components inventory Milestone 23); brownfields can be categorised under ‘abandoned/ruderal land’. We have discussed that in detail in our first paper (Chowdhury et al. 2019) and how the contamination concern of such lands makes it difficult for them to be used in their full potential as greenspace. Some clarification is added in the main body of text (L84) as not to cross the limit of the abstract word limit.

These lines are added (L 73 – 79 in the revised manuscript):

‘Brownfields in their present state can be considered as an UGS, ‘abandoned/ruderal land’ that supports spontaneous vegetation based on the UGS typology by Haase et al. (2015). Though brownfields can provide ample of regulating and supporting ecosystem services, their use is unregulated and limited largely due to potential contamination from previous exploitations (Mathey et al., 2018; Mathey & Rink, 2010). Retrofitting brownfields as more usable greenspaces can add on to the cultural ecosystem services as well as increasing the environmental benefits (Mathey et al., 2018).’

 L32: Unclear what is meant with the “realisation of a greenspace”. Does this mean to free up space in the competitive urban environment or does it mean to convert brownfields to parks? The latter would depend on the definition of what is considered “greenspace”. 

 L84: Also brownfields already provide some important ecosystem services. Would be good to be more specific here, what kind of improvements derived from greenspaces are considered in the study. 

 L34(ff): What do you mean with “a modified method”? Which method and what kind of modification? The abstract is edited, and this referenced line has been taken out. 

 L36: Is "the site" the case study introduced in the next sentence or is it meant more generally? The objectives are reformulated and now it says, ‘UGS development’ instead of ‘site development. It was meant generally. 

 L38: We are approaching the end of the abstract and this is still methods description. I would suggest to structure the abstract more or less equally into introduction, methods, results and discussion/conclusions. We have taken this consideration in rewriting the abstract of the revised manuscript (L 24 – 45):

Abstract 

Urban greenspaces (UGS) provide a range of ecosystem services and are instrumental in ensuring the liveability of cities. Whilst incorporating UGS in increasingly denser cities is a challenge to planners, brownfields form a latent resource with the potential of being converted into UGS. Transformation of brownfields to greenspaces, however, requires engagement of a variety of stakeholders, from providers to users. The overall aim of this study was to support effective and realistic realisations of UGS in the context of urban brownfields’ regeneration and stakeholder engagement. A working process was developed to: 1) integrate methods relevant for UGS realisation for a) identification and categorising of relevant stakeholders, b) mapping their interests and resources, c) identifying various challenges, and d) matching those challenges with the mapped resources over the timeline of UGS development; and 2) apply these methods to assess relevance and shortcomings. The methods were applied to a study site in Sweden, and data was collected using a questionnaire survey. The respondents’ comments indicated that the combination of several uses, especially integrated with an urban park, is preferable. Visualisation was an important component for data analysis: stakeholder categorisation was effectively visualised using a Venn diagram, and the needed mobilisation of resources among stakeholders to manage identified challenges was visualised using a timeline. The analysis demonstrates the need for collaboration between stakeholders to achieve an effective realisation of UGS and how multiple methods can be used in concert to map stakeholders, preferences, challenges, and resources for a particular site. The application at a study site provided site-specific data but the developed stakeholder categorisation, and the method for matching identified challenges with the stakeholders’ resources using a timeline, can be generalised to applications at other sites.

Keywords: Urban greenspaces (UGS), brownfields, stakeholder analysis

 L41: Unclear why this sentence starts with “however”, is there a contradiction with the previous sentence? As pointed out, the word ‘however’ serves no additional purpose here and thus is removed.

 L89-90: I don't understand why this sentence is needed. In the next paragraph you state that identifying and categorizing stakeholders is the main goal of this paper? Agreed, this line is deleted.

We have also revised the aim. 

 L113: So, is the methods development the purpose of the paper or is the purpose the identification and categorization of stakeholders in a case study? This difference is important, as the method should follow the purpose and not the other way around. The aim and objectives are revised as explained in the first row. Hopefully the division is more distinct and clearer in the reworked ones. 

 L127: Better to refer to a peer-reviewed reference for the Gigamapping approach This reference is added (L 144 of the revised manuscript): 

Davidová, M., & Zímová, K. (2021). Colreg: The tokenised cross‐species multicentred regenerative region co‐creation. Sustainability (Switzerland), 13(12). https://doi.org/10.3390/SU13126638

 L129: Should “thus” be replaced by “then”? Agreed, done.

 L131: Maybe state already what kind of method was new Method section is reworked, and this line has been removed. 

 L137-139: I think here it will be enough to just report what has been done for what reason rather than listing all methodological options that were considered We want to list all the data collection methods considered because the method we end up using is not the suggested data collection method for the tools we are using for stakeholder analysis (e.g Crosby method). We wanted to present our reasonings and limitations. 

 L145: What is a scoping review and how is it carried out? A reference is added (in the manuscript as well as down below) to explain this type of review. It is mostly a preliminary assessment to find literature within the scope of the present study. 

Grant, M. J., & Booth, A. (2009). A typology of reviews: An analysis of 14 review types and associated methodologies. Health Information and Libraries Journal, 26(2), 91–108. https://doi.org/10.1111/j.1471-1842.2009.00848.x

 L155: Looking at the methods selection part in the SI (as Fig 2 is not legible) I do not understand why there is so much consideration of different methods if the Gigamapping tool seems to already have been decided on. Fig 2 of the previous manuscript has been deleted. Also, the selection among different methods have been deleted from the manuscript and has been limited to the supplementary material (gigamap). 

 L171: Who is the research team and how is their expertise suitable or representative for the overall process that you are trying to analyze? ‘Research team’ is changed to ‘first author to collaborate with the other authors’.

 L176: It may not be enough to just mention the challenge of simplifying the gigamap, rather there should be some explanation how you dealt with this challenge. As well as the objectives have been revised, stress on gigamapping has been reduced in the revised manuscript. 

However, it was not deemed reasonable to take out completely since the Gigamapping actually has helped with the analysis of the collected data as we elaborated on the discussion (L 558 – 561). 

 L182: I agree that the gigamap is complex, but how does that make it useful to help solve the problem or reach the objectives of the study? 

 L266: There should be some consideration how far the number of respondents allows answering the questions. I do not see an issue with following a more qualitative approach and using the surveys as a proof-of-concept. Yet they may not be sufficient to quantitatively compare different groups of stakeholders. Thank you for your suggestion. The study is performed based on what you have suggested, using the survey responses as ‘proof of concept’ and the analysis is rather qualitative even for the more quantitative responses (preference of UGS). We think this consideration is apparent in the way the result is presented in the revised manuscript.

To address what is missing or the limitation, we have added these lines in the discussion (L 597 – 602): ‘The survey received quite a limited number of responses, although the responses given did provide a lot of information. The Recycling and Water Office was acknowledged as an important stakeholder by other respondents, especially for the bioswale option, but the office unfortunately did not respond to the survey. The Environmental Department at the municipality have different functions, and it is a bit uncertain whether e.g. the part that acts as controlling authority, is represented. They could also have given input regarding the view on any constraints a potential contamination situation would pose.’

 L272: What about the remaining participants? Given the lack of balance between sampled groups, a comparison between stakeholder groups does not seem possible. Maybe best to lump all responses together as an indicator for general public interest? The ratio of respondents’ type can be considered proportionate given 6 respondents are from the ocal governments and 18 being local residents. This difference plays a part in preference, resources they could bring, etc. later on in the result section. So, the division is kept. 

 L304: Do you mean "not all stakeholders could be categorized"? Thank you, the sentence (L 273 in the revised manuscript) has been edited accordingly. 

 L344: Fig. 4 looks nice but it does not look like a method, rather like a result. Also, it remains unclear from the methods, how this diagram has been developed and why you think it is conclusive Fig. 4 (Fig 3 in the revised manuscript) is a result of application of the stakeholder categorisation method. We have clarified that in the figure.

 L321-382: This whole section is a very detailed description of the stakeholders in the case area and it is unclear, how this is an outcome of the methodology or supports the use of the method. We have revised the aim and objectives and thus, the result section is heavily revised as well. So hopefully, these concerns addressed here in the comments may are dealt with indirectly in the revised manuscript (L ).

 L399: What are medium answers? It was meant that the respondents showed average interest. The wording is changed to ‘average’.

 L414: If there is no trade-off or cost involved in the decision, of course everyone would opt for a new park if asked. We wanted to show what is known as well, there are some restrictions as you have pointed out, trade-offs, that come in later when the challenges are identified. We suppose nothing is actually expected to be changed regarding this line. 

 L418: This already sounds like part of the discussion We consider this to be result though, as in how the respondents view different greenspaces and what do they think of how they would use them. 

 L426: The painted pictures in Fig. 5 are very nice. If you have the rights to use them it would be good to make them much bigger in the Fig. to show some intuitive images to the reader Thank you very much! The images are created by the first author and thus, we have the rights to use them. The image (Fig 4 in the revised manuscript) is adjusted following the comments. 

 L427-469: This section is a description of different possible types of UGS in the study area, which is again not a result of the methodology applied. The explanation of the greenspaces here are the excerpt from the responses of the survey questionnaire. The idea here was to understand how the respondents perceive certain greenspaces and how they reason their preferences. So, this is the output of the questionnaire survey and thus, belongs to result section. 

 L505-506: what is this categorization based on? The basis of the challenge mapping in the cited literature has been clarified. Edited sentence:

The challenge categorisation proposed by Fernandes et al. (2020) to map stakeholders’ perception on challenges on brownfield redeployment includes six groups: governance, infrastructure, territory, finance, culture, and environment.

 L607: Fig. 8 is almost impossible to read and understand The figure (Figure 7 in the revised manuscript) has been revised to increase visibility.

 L650-666: This whole paragraph is on methods selection and not sure how far it fits in the discussion (given the current results section) This paragraph has been removed entirely in the revised manuscript. Instead, another has been added that discusses the limitation of the study (e.g. stakeholders outside of the study scope, remediation, etc.) (L 627 – 656 in the revised manuscript). The paragraph is added in the Reviewer 1 comment table (Row 1, column 3)

Reviewer 3

This innovative study aims to help facilitate the redevelopment of urban brownfields into urban green spaces (UGS) in urban metro areas by using literature review, mapping, and data analysis to 1) identify and characterize stakeholders; 2) map stakeholder interests and resources; 3) map challenges to brownfield redevelopment; and 4) match challenges with resources. The study draws case-specific conclusions about the effectiveness of each aforementioned method in n Polstjärnegatan, Gothenburg (SE). This paper has great potential to give important insights to brownfield redevelopment and greenspace development efforts. Below I outline some suggestions for consideration to improve the quality and impact of the manuscript.

Table 4. Measures and responses to reviewer 3’s comments. The reviewer’s reference to lines in the manuscript are to the original manuscript. 

 Comments How it is addressed (mostly comments are descriptive and does not require action, so those were marked ‘partially’ as well)

Introduction I suggest updating your references on the influence or urban greenspace on mental and physical health. Your most recent reference (in line 49) is 2010. Below are some good empirical articles. The revised manuscript has some updated reference. But the references you have suggested were not visible, so we were unfortunately not able to add the ones specified by the reviewer. 

 You briefly mention that brownfields are potentially contaminated. Yes, if not proven otherwise, all brownfields are considered to be potentially contaminated. We understand not discussing this aspect is a limitation of this study, but this has been done intentionally. A paragraph on the result section is added in the revised manuscript that discusses this topic in detail (see reply to Reviewer 1 comment table (Row 1, column 3))

 In line 98, do you mean members of the public or the community? “Members of the society” can have a very broad meaning. I consider everyone, everything, and every organization as a part of society. Be more specific here. Since what is considered ‘society’ (e.g. residents, community-based organisations and academic institutions) has been expressed just previously in the manuscript (L 104 in the revised manuscript), maybe repeating that distinction wont be necessary. 

Methodology It is unclear in the description of stakeholder analysis whether the UGS that respondents selected as ‘interest’ where actual existing UGS or possible locations for UGS/former brownfields. Authors should be more explicit in describing this question/process in lines 210-220. This has been now added in the location suggested in the revised manuscript. The line now states (L 182 – 183 in the revised manuscript), ‘An ‘interest’ was interpreted as a stakeholders’ preference for an UGS to be developed on the study site which is presently in derelict condition with sparse vegetation (more detail on section 2.4).’

 Authors should further describe the reference group that was given the survey initially. See lines 252-254 “The online data collection process started with the members of a reference group to the research project…… who were most municipality representatives.” That would have been preferred by us, the authors, as well. But it is difficult to do so without risking revealing the identity of the stakeholders or the project or us , which would not be in line with the journal’s double blind peer review system. We have tried to address this by mentioning them in the acknowledgement which is unfortunately not part of the main manuscript. 

 Did the authors collect any other demographics from the sample (e.g., race/ethnicity, age, # of years in residence, gender, education)? Factors such as these might be important for understanding responses, interests, and challenges. That would have been an interesting study indeed, however we have not considered that in this study. 

Results Here in the results the reader finally gets a description of UGS interests. These are the types of UGS that the stakeholder prefers. A brief description of the type options should be placed in the methodology section. (See first bullet point under methodology) In the section 2.3 Data Collection, it states (L 212 - 213), ‘The list of possible UGS at the study site and description of these UGS in the questionnaire was based on Chowdhury et al. (2020).’ The cited literature gives a detail description of all the greenspaces used in the questionnaire. 

Discussion I notice that there is not much discussion of contamination of the brownfield and the health impacts of its redevelopment into UGS. Authors should at least discuss this and perhaps why this didn’t present itself as much in the “challenges” for stakeholders. Are stakeholders aware of potential exposure and risks to health? We understand not discussing this aspect is a limitation of this study, but this has been done intentionally. A paragraph on the discussion section (L 627 – 656 in the revised manuscript) is added in the revised manuscript that discusses this topic in detail (see reply to Reviewer 1 comment table (Row 1, column 3))

 Perhaps racial, gender, and socioeconomic dynamics do not play as large a role in the local contexts (Sweden) as it does in U.S. contexts. As the authors note that their findings can and should be generalizable to broader contexts involving urban brownfield redevelopment into greenspace, it might be worthwhile to at least discuss how different social contexts could shape findings. A cross Atlantic study would honestly be very interesting considering the difference in society. Even within the city chosen here (Gothenburg), if we chose a different locality, these issues identified may have played a significant role. This line is in the discussion (L 657 - 666): The working process developed in this study for identifying and categorising stakeholders and their interests, challenges, and resources, can facilitate effective and realistic realisations of UGS in the context of regenerating urban brownfield sites. It demonstrates how multiple methods can be used in concert to map stakeholders, preferences, challenges, and resources for a particular site. For another site, other methods may be more suitable depending on the particular setting. Specifically, the Crosby method was found to be limiting for this site, although the survey did provide more in-depth information in the open-ended questions. The demonstration at the study site provided site-specific data but, the developed stakeholder categories and groups visualised in the Venn diagram, as well as the timeline of mobilisation of the resources of stakeholders to manage the challenges of realising an UGS, can be generalised to applications at other sites in Sweden and beyond.

Editorial review

Journal requirements 

Done.

Please include in the Methods section of your manuscript text the following:

a) A brief statement regarding exemption from ethical review for this study under Swedish law, including the link to the relevant legislation you have provided in the Ethics Statement of the online submission information.

b) A brief statement regarding participant consent as noted in the Ethics Statement of the online submission information. Additionally, for the verbal consent obtained, please state how the consent was documented and witnessed. The following statements are added in the method section

An ethical review was not required as the content of the questionnaire survey for this study is exempt from any such requirement according to the Swedish data protection act (Lag (2018:218), section 3 and 4).

The online questionnaire survey was performed with explicit consent of the participants which was required to move forward with filling the survey. For offline data collection, the respondents were also first asked to consent to the purpose of the study and the survey was filled only after the verbal consent was given. Both online and offline survey participants could choose to stay anonymous. 

Thank you for stating the following in the Acknowledgments Section of your manuscript: 

[This work is supported by Formas (Grant number: 2017-00246).]

 [YES

This work is supported by Formas (Grant number: 2017-00246)]

Please include your amended statements within your cover letter; we will change the online submission form on your behalf. The edited cover letter currently consists of the statement: This work is supported by Formas (Grant number: 2017-00246).

And that statement is removed from the manuscript and changed to: This work is supported by Formas.

The funding statement should still read: 

 [YES

This work is supported by Formas (Grant number: 2017-00246)]

Thank you for stating the following in your Competing Interests section: 

[NO

The authors declare that they have no known competing financial interests or personal relationships that could have appeared to influence the work reported in this paper].

 This information should be included in your cover letter; we will change the online submission form on your behalf. The edited cover letter currently consists of the statement: The authors have declared that no competing interests exist. 

5. We note that Figure 3 in your submission contain map/satellite images which may be copyrighted. All PLOS content is published under the Creative Commons Attribution License (CC BY 4.0), which means that the manuscript, images, and Supporting Information files will be freely available online, and any third party is permitted to access, download, copy, distribute, and use these materials in any way, even commercially, with proper attribution. For these reasons, we cannot publish previously copyrighted maps or satellite images created using proprietary data, such as Google software (Google Maps, Street View, and Earth). For more information, see our copyright guidelines: http://journals.plos.org/plosone/s/licenses-and-copyright.

a) You may seek permission from the original copyright holder of Figure 3 to publish the content specifically under the CC BY 4.0 license. 

The figure 3 (presently figure 1) has been reworked with satellite images and maps from sources that adheres to the CC BY 4.0 guideline.

Current source is Lantmäteriet for the images, and according to the documentation provided, they support reproduction via CC BY 4.0.

Excerpt from the document which can be accessed here (in Swedish):

‘If copyright is claimed by the data provider, you must also indicate this by the symbol © and the name of the relevant authority, for example © Lantmäteriet. The copyright marking is intended to inform the reader that the amount of data is covered by copyright. This means that further use and dissemination of the amount of data to third parties usually requires permission from or agreement with the person claiming copyright. Lantmäteriet, the Swedish Maritime Administration, the Swedish Geological Survey (SGU) and Statistics Sweden (SCB) claim copyright. When you publish or otherwise distribute data in accordance with the applicable terms of use from these authorities, you must state © the Authority. This also applies to the data of many other authorities and organizations, and may also apply to the data that is provided free of charge or as so-called open data. What applies is stated in the license terms that belong to the respective amount of data. You will find the conditions in the metadata for the data set (the symbol or when you search in the Geodata portal) under the Restrictions tab. It says e.g. “Access Restrictions: Copyright; Usability Restrictions: Creative Commons Attribution (CC BY, version 4.0)”. If you write © The Authority, it is an approved acknowledgment of copyright.’

Current heading of the figure:

Figure 1. Top: Location map integrated with the concept plan of the Karlastaden development project with the site area highlighted (Karlastaden plan redrawn from Göteborg stad (2017); base map and orthophoto ©Lantmäteriet, used under Creative Commons License CC BY 4.0). Bottom: Physical conditions of the site as of January 11, 2020 (source: first author). 

References

Chowdhury, S. (2020). An Assessment of the Potential for Bio-based Land Uses on Urban Brownfields [Licentiate thesis, Chalmers University of Technology]. https://research.chalmers.se/en/publication/520006

Chowdhury, S., Kain, J.-H., Adelfio, M., Volchko, Y., & Norrman, J. (2020). Greening the Browns: A Bio-Based Land Use Framework for Analysing the Potential of Urban Brownfields in an Urban Circular Economy. Sustainability, 12(15), 6278. https://doi.org/10.3390/su12156278

Drenning, P., Chowdhury, S., Volchko, Y., Rosén, L., Andersson-Sköld, Y., & Norrman, J. (2022). A risk management framework for Gentle Remediation Options (GRO). Science of the Total Environment, 802, 149880. https://doi.org/10.1016/j.scitotenv.2021.149880

Fernandes, A., Figueira de Sousa, J., Costa, J. P., & Neves, B. (2020). Mapping stakeholder perception on the challenges of brownfield sites’ redevelopment in waterfronts: the Tagus Estuary. European Planning Studies, 28(12), 2447–2464. https://doi.org/10.1080/09654313.2020.1722985

Göteborg stad. (2017). Detajlplan för bostader och verksamheter vid karlavagnsplatsen inom stadsdelen Lindholmen i Göteborg.

Haaland, C., & van den Bosch, C. K. (2015). Challenges and strategies for urban green-space planning in cities undergoing densification: A review. Urban Forestry and Urban Greening, 14(4), 760–771. https://doi.org/10.1016/j.ufug.2015.07.009

Haase, D., Kabisch, N., Strohbach, M., Eler, K., & Pintar, M. (2015). Urban GI Components Inventory Milestone 23 (Vol. 7).

Harvey, D. (2006). Neo-Liberalism as Creative Destruction. In Geografiska Annaler. Series B, Human Geography (Vol. 88, Issue 2).

Mathey, J., Arndt, T., Banse, J., & Rink, D. (2018). Public perception of spontaneous vegetation on brownfields in urban areas—Results from surveys in Dresden and Leipzig (Germany). Urban Forestry & Urban Greening, 29, 384–392. https://doi.org/10.1016/J.UFUG.2016.10.007

Mathey, J., & Rink, D. (2010). Urban Wastelands - A Chance for Biodiversity in Cities? Ecological Aspects, Social Perceptions and Acceptance of Wilderness by Residents. Urban Biodiversity and Design, 406–424. https://doi.org/10.1002/9781444318654.CH21

Lag (2018:218), (2018). https://rkrattsbaser.gov.se/sfst?bet=2018:218

Roy, P. (2011). Non-profit and Community-based Green Space Production in Milwaukee: Maintaining a Counter-weight within Neo-liberal Urban Environmental Governance. Http://Dx.Doi.Org/10.1080/13562576.2011.625220, 15(2), 87–105. https://doi.org/10.1080/13562576.2011.625220

---

## [Decision Letter · Decision Letter 1]

23 Aug 2022

PONE-D-22-04756R1Transforming brownfields into urban greenspaces: A working process for stakeholder analysisPLOS ONE

Dear Dr. Chowdhury,

Thank you for submitting your manuscript to PLOS ONE. After careful consideration, we feel that it has merit but does not fully meet PLOS ONE’s publication criteria as it currently stands. Therefore, we invite you to submit a revised version of the manuscript that addresses the points raised during the review process.

In your response to the reviewer comments, please be sure to address the minor issues brought up by reviewer #2 as well as the more significant issues about the questionnaire pointed out by reviewer #4.

We look forward to receiving your revised manuscript.

Kind regards,

Theodore Raymond Muth

Academic Editor

PLOS ONE

Reviewers' comments:

Reviewer's Responses to Questions

**Comments to the Author**

1. If the authors have adequately addressed your comments raised in a previous round of review and you feel that this manuscript is now acceptable for publication, you may indicate that here to bypass the “Comments to the Author” section, enter your conflict of interest statement in the “Confidential to Editor” section, and submit your "Accept" recommendation.

Reviewer #2: All comments have been addressed

Reviewer #4: (No Response)

2. Is the manuscript technically sound, and do the data support the conclusions?

Reviewer #2: Yes

Reviewer #4: Partly

3. Has the statistical analysis been performed appropriately and rigorously? 

Reviewer #2: N/A

Reviewer #4: N/A

4. Have the authors made all data underlying the findings in their manuscript fully available?

Reviewer #2: No

Reviewer #4: Yes

5. Is the manuscript presented in an intelligible fashion and written in standard English?

Reviewer #2: Yes

Reviewer #4: Yes

6. Review Comments to the Author

Reviewer #2: I carefully went through the manuscript “Transforming brownfields into urban greenspaces: A working process for stakeholder analysis” a second time and can confirm that the text has improved significantly. Most comments have been addressed in a very thorough manner and the authors have done a good job in reframing the text and reworking the figures which are now much clearer and pleasant. Especially the research questions and the methodology are now more concise and link better with the results of the study.

However, I still suggest further work on the text, mostly in the results section. I acknowledge that the manuscript has been shortened as compared to the earlier version, but it is still unreasonably long, especially given the limited actual evidence in terms of respondents in the survey. The kindly provided tracked changes reveal that relatively little has changed in the results section compared to the earlier version. Therefore, I again suggest some shortening, especially in sections 3.1 and 3.2. These parts are very descriptive and the contents do not justify such a length that is hard to follow for someone who is not very familiar with this particular case study. Interestingly, the Discussion section now includes a well-written bulleted section (l. 571 ff) summarizing the “main insights”, which is essentially a duplication of the results. I suggest removing this concise summary of the results from the discussion and merging it with the actual results section. This process of rewording should help shortening the results significantly which will help making the overall paper more accessible and meaningful. Instead of describing all stakeholder with all detail, I would rather suggest using more actual quotes from the qualitative part of the surveys, which could help making the text more engaging to read.

A few minor comments on wording:

L 32: challenges with greenspace implementation?

L 35: add type of questionnaire (qualitative, quantitative) and number of respondents

L 37 : remove «effectively»

L76 : remove “of”

L 76: I guess there are other limitations than just contamination (accessibility, ownership, …) so maybe replace “largely” with “for example”

L 127: For consistency change wording to “identifying and categorising” or “identification and categorisation”

L154-155: Not sure if this last sentence is needed.

Fig 3 and elsewhere: I am a bit unsure about the “Everyone” category. Can this category be renamed to what can actually be considered the full population relevant for this study? Maybe the inhabitants of the city or region?

Reviewer #4: This article develops an integrated stakeholder analysis method for urban green space realisation from urban brownfield using a visualisation tool. The application of method is demonstrated by a case study Polstjärnegatan in Gothenburg. This is an interesting topic to explore given the strong demand for green spaces after the pandemic.

Major comments:

- The authors could justify the advantage and disadvantage of applying the questionnaire survey as the stakeholder analysis method compared to other methods like interview, focus group discussion and workshop, as it is a new approach applied to understand stakeholder interests (Section 2.3, line 199-204). For example, questionnaire is potentially a useful strategy to collect different views for regeneration cases.

- A total of 31 responses are collected, while 18 of them are from local students and local residents. Students could be important stakeholder but they may not be long-term residents in the area, so I wonder if they could represent the community opinion. The feedbacks from other stakeholders (like local business) are limited and maybe not enough to show the interest of whole stakeholder group. It would be helpful to add a bit more site context about the general local community component and distribution (e.g. local demographics, how many residence estates vs. student facilities). Otherwise it would be hard to determine whether the collected samples are representative enough.

- I wonder how identified challenges in this study are associated with broader context of brownfield-UGS regeneration in the country. Conflicts of interest (mentioned in line 113 in introduction) seems to be a key challenge discussed in the literature, while the selected case study does not directly demonstrate this potential challenge. The discussion on what insights can be learned from the case study to inform similar cases in the future seems to be insufficient (Section 3.3 and line 589-592 in Section 4).

Minor comments:

- Line 213: can add a bit more detail about the ten UGS types used in the questionnaire (for example, with a reference to UGS classification), and describe other questions enquired in the questionnaire a bit (1-2 sentences).

- Figure 7 (bottom). ‘Human resourcesin the form of work neighbours…’ needs a space between ‘resources’ and ‘in’.

7. PLOS authors have the option to publish the peer review history of their article (what does this mean?). If published, this will include your full peer review and any attached files.

Reviewer #2: No

Reviewer #4: No

---

## [Author Response · Author response to Decision Letter 1]

6 Oct 2022

PONE-D-22-04756

Transforming brownfields into urban greenspaces: A working process for stakeholder analysis (revised from: Transforming brownfields into urban greenspaces: Development and application of a multi-method approach for Stakeholder analysis)

PLOS-ONE

Dear Editor,

We have revised the article PONE-D-22-04756 in accordance with the changes suggested by the editor and the reviewers. We have explicitly addressed the specific comments of the reviewers, see details in Table 1. The comments were constructive and helped to improve the manuscript. 

Best regards,

The authors

Editor’s summary points

In your response to the reviewer comments, please be sure to address the minor issues brought up by reviewer #2 as well as the more significant issues about the questionnaire pointed out by reviewer #4.

Reviewers’ comments

Reviewer 2

I carefully went through the manuscript “Transforming brownfields into urban greenspaces: A working process for stakeholder analysis” a second time and can confirm that the text has improved significantly. Most comments have been addressed in a very thorough manner and the authors have done a good job in reframing the text and reworking the figures which are now much clearer and pleasant. Especially the research questions and the methodology are now more concise and link better with the results of the study.

General response:

Thank you for your insights and reflections on the revised texts. Your comments on the original manuscript have been fundamental in restructuring the text and we are very grateful for such through guidance.

However, I still suggest further work on the text, mostly in the results section. I acknowledge that the manuscript has been shortened as compared to the earlier version, but it is still unreasonably long, especially given the limited actual evidence in terms of respondents in the survey. The kindly provided tracked changes reveal that relatively little has changed in the results section compared to the earlier version. Therefore, I again suggest some shortening, especially in sections 3.1 and 3.2. These parts are very descriptive, and the contents do not justify such a length that is hard to follow for someone who is not very familiar with this particular case study. Interestingly, the Discussion section now includes a well-written bulleted section (l. 571 ff) summarizing the “main insights”, which is essentially a duplication of the results. I suggest removing this concise summary of the results from the discussion and merging it with the actual results section. This process of rewording should help shortening the results significantly which will help making the overall paper more accessible and meaningful. Instead of describing all stakeholder with all detail, I would rather suggest using more actual quotes from the qualitative part of the surveys, which could help making the text more engaging to read.

Response

We appreciate the constructive feedback, and we also acknowledge that the length of the paper might be challenging. Still, as the article reports on a composite methodology and its application on a case, we feel that the length is justified, and we consider the narrative and descriptive sections as part of the “storytelling” of the paper. In consequence, we also believe that summarizing the main insights in the discussion is helpful for those not delving into the whole paper. Therefore, we have chosen not to follow this suggestion. 

L 32: challenges with greenspace implementation?

Response: Added as suggested.

L 35: add type of questionnaire (qualitative, quantitative) and number of respondents

Response: The questionnaire has both qualitative or open ended (e.g. Q3 Explain briefly why you are interested in the selected urban greenspace) and quantitative or close ended (e.g. Q4 Please rate your interest in the selected urban greenspaces using the following numbers: 0 = no interest; 1 = low; 2 = medium; 3= high). So, fixing a type wouldn’t do the questionnaire justice and both are added. The number of respondents is added. The sentences are modified: ‘The methods were applied to a study site in Sweden, and data was collected using a questionnaire survey with both open and close-ended questions. The survey received 31 responses and the respondents’ comments indicated that the combination of several uses…’

L 37: remove «effectively»

Response: Removed as suggested.

L76: remove “of” 

Response: Removed as suggested.

L 76: I guess there are other limitations than just contamination (accessibility, ownership, …) so maybe replace “largely” with “for example” 

Response: Edited as suggested.

L 127: For consistency change wording to “identifying and categorising” or “identification and categorisation” 

Response: Edited as suggested.

L154-155: Not sure if this last sentence is needed. 

Response: Has been changed into: The full complexity of the final Gigamap produced in this study can be appreciated in the attached supplementary material 1. 

Fig 3 and elsewhere: I am a bit unsure about the “Everyone” category. Can this category be renamed to what can actually be considered the full population relevant for this study? Maybe the inhabitants of the city or region? 

Response: The category is named reflecting on the respondents’ comments. In a frank way, several respondents explained how they think ‘everyone’ should be involved in realising UGS and we want to capture that essence. But we understand the need for further information and the this line in the text clarifies (L 305-307): The proposed categorisation has one main category (everyone) which then includes three cluster categories (government, local community, non-local community/visitors), where all stakeholders that are relevant for this study can appear in one or more clusters.

Reviewer 4

This article develops an integrated stakeholder analysis method for urban green space realisation from urban brownfield using a visualisation tool. The application of method is demonstrated by a case study Polstjärnegatan in Gothenburg. This is an interesting topic to explore given the strong demand for green spaces after the pandemic.

The authors could justify the advantage and disadvantage of applying the questionnaire survey as the stakeholder analysis method compared to other methods like interview, focus group discussion and workshop, as it is a new approach applied to understand stakeholder interests (Section 2.3, line 199-204). For example, questionnaire is potentially a useful strategy to collect different views for regeneration cases 

Response: The following sentence has been added: A questionnaire survey for data collection also meant that respondents could answer at their own convenience and being largely online and anonymous, helped the respondents to be frank with their answers (Williamson, 2018).

A total of 31 responses are collected, while 18 of them are from local students and local residents. Students could be important stakeholder but they may not be long-term residents in the area, so I wonder if they could represent the community opinion. The feedbacks from other stakeholders (like local business) are limited and maybe not enough to show the interest of whole stakeholder group. It would be helpful to add a bit more site context about the general local community component and distribution (e.g. local demographics, how many residence estates vs. student facilities). Otherwise, it would be hard to determine whether the collected samples are representative enough. 

Response: Students form a large part if the inhabitants in this area and will continue to do so. To make this clear, the following text has been re-formulated: 

The study site at Polstjärnegatan is located within the Lindholmen district in Gothenburg (Sweden). Population of Lindholmen is around 5000 according to 2022 census but more than 10000 students come to study here due to the district being an education hub with several schools and a campus of Chalmers University of Technology (Göteborgs stadsledningskontor, 2022; Lundby stadsdelsförvaltning, 2020). Lindholmen district has less than 3000 housing units at present and a significant part of them are student apartments (nearly 700 registered apartments by the student housing associations alone) (Göteborgs stadsledningskontor, 2022; SGS Studentbostäder, 2022). The site is part of the concept plan of Karlastaden, a large-scale housing and commercial facility (Fig. 2, top), whose construction (as of 2022) is in progress (Göteborgs Stad, 2017, 2022). The development is projected to finish by 2026 and will add 2000 more apartments to the Lindholmen area (Göteborgs Stad, 2017, 2022). The site has confirmed contamination issues due to previous uses (Kaltin & Almqvist, 2016). It is surrounded by roads and a railway on all sides: Polstjärnegatan to the south, Karlavagnsgatan to the east, a petrol station and a fast-food restaurant to the west, and a railway (Hamnbanan) as well as a motorway (Lundbyleden) to the north (see Fig. 2). 

I wonder how identified challenges in this study are associated with broader context of brownfield-UGS regeneration in the country. Conflicts of interest (mentioned in line 113 in introduction) seems to be a key challenge discussed in the literature, while the selected case study does not directly demonstrate this potential challenge. The discussion on what insights can be learned from the case study to inform similar cases in the future seems to be insufficient (Section 3.3 and line 589-592 in Section 4). 

Response: Thank you for pointing this out. The following paragraph has been added in the discussion:

In a review on the challenges associated with UGS planning in cities, Haaland & van den Bosch (2015) confirms that densification processes such as infill development and consolidation can pose a threat to UGS. The site in this study is in a rapidly developing and densifying urban district and is part of a development project, Karlastaden, that consists of eight urban blocks of mixed commercial and residential development (cite). Even though the site is designated to be designed as a greenspace, it can be considered as part of a larger urban densification project. With the speed of urbanisation in the area, the concern of the local stakeholders (Challenge S1 and S2 in Fig 6) that greenspaces in the area can come under more pressure to be repurposed for more economically beneficial (e.g. housing or commercial) land use. The dense Karlastaden comes with another challenge is that it consists of four high-rises and impact of such in planning the greenspace. Although unique to the local context, high-rise neighbourhoods exist in many cities across the world and the design challenges are studied for different contexts (FFLA, 2018; Ye et al., 2020). There are now more subtler challenges, as explained by Colding et al. (2020), that can conjugate and pose a greater threat in realising and maintaining UGS in cities. Challenges regarding ensuring maintenance (Challenge G3 in Fig 6) that stems from lack of financial resources (Challenge F1 and F2) can potentially lead to outsourcing the management to private sector. Such challenges as observed by the stakeholders at the study site are argued by Colding et al. (2020) to result in gradual loss of access to greenspaces across many cities due to privatisation. This can negatively affect the urban inhabitants, specifically for citizens in Swedish cities, who largely associate their well-being with access to natural areas (Elbakidze et al., 2017) and make frequent visits to different greenspaces located within the cities (Rydberg & Falck, 1998). Even with almost 70% of land area covered with forest in Sweden (SLU, 2022), more than 50% of forest visits happen in the urban forests with most frequent visits occurring in the ones located withing 1 km of the residents’ home (Kardell, 1985; Rydberg & Falck, 1998). At the study site, the respondents also emphasised on the importance of UGS by reasoning how stakeholders across all domains potentially benefit from them. The challenges identified for UGS realisation by the respondents as well as the resources they bring to tackle the challenges can help to support UGS realisation and maintenance in cities. 

Line 213: can add a bit more detail about the ten UGS types used in the questionnaire (for example, with a reference to UGS classification), and describe other questions enquired in the questionnaire a bit (1-2 sentences). 

Response: These lines are added: ‘The list of possible UGS at the study site and description of these UGS in the questionnaire was based on Chowdhury et al. (2020), using the classification/categorisation of the Greensurge project (Haase et al. 2015). The UGS presented in the questionnaire were: bioswale, urban park, grassland/shrubland, meadow orchard, allotment, community garden, commercial agriculture, biofuel production, and spontaneous vegetation.’

 Figure 7 (bottom). ‘Human resources in the form of work neighbours…’ needs a space between ‘resources’ and ‘in’. 

Response: Done

---

## [Decision Letter · Decision Letter 2]

23 Nov 2022

Transforming brownfields into urban greenspaces: A working process for stakeholder analysis

PONE-D-22-04756R2

Dear Dr. Chowdhury,

We’re pleased to inform you that your manuscript has been judged scientifically suitable for publication and will be formally accepted for publication once it meets all outstanding technical requirements.

Kind regards,

Theodore Raymond Muth

Academic Editor

PLOS ONE

Additional Editor Comments (optional):

Reviewers' comments:

Reviewer's Responses to Questions

**Comments to the Author**

1. If the authors have adequately addressed your comments raised in a previous round of review and you feel that this manuscript is now acceptable for publication, you may indicate that here to bypass the “Comments to the Author” section, enter your conflict of interest statement in the “Confidential to Editor” section, and submit your "Accept" recommendation.

Reviewer #2: All comments have been addressed

Reviewer #4: All comments have been addressed

2. Is the manuscript technically sound, and do the data support the conclusions?

Reviewer #2: Yes

Reviewer #4: Yes

3. Has the statistical analysis been performed appropriately and rigorously? 

Reviewer #2: N/A

Reviewer #4: N/A

4. Have the authors made all data underlying the findings in their manuscript fully available?

Reviewer #2: Yes

Reviewer #4: Yes

5. Is the manuscript presented in an intelligible fashion and written in standard English?

Reviewer #2: Yes

Reviewer #4: Yes

6. Review Comments to the Author

Reviewer #2: (No Response)

Reviewer #4: Authors have addressed the comments in the last round. No further comments here.

7. PLOS authors have the option to publish the peer review history of their article (what does this mean?). If published, this will include your full peer review and any attached files.

Reviewer #2: **Yes: **Fritz Kleinschroth

Reviewer #4: No

---

## [Editor Report · Acceptance letter]

26 Dec 2022

PONE-D-22-04756R2 

Transforming brownfields into urban greenspaces: A working process for stakeholder analysis 

Dear Dr. Chowdhury:

I'm pleased to inform you that your manuscript has been deemed suitable for publication in PLOS ONE. Congratulations! Your manuscript is now with our production department. 

Kind regards, 

on behalf of

Dr. Theodore Raymond Muth 

Academic Editor

PLOS ONE